# RUNX1/RUNX1T1 mediates alternative splicing and reorganises the transcriptional landscape in leukemia

Vasily V. Grinev [1✉], Farnaz Barneh[2], Ilya M. Ilyushonak[1], Sirintra Nakjang[3], Job Smink[2], Anita van Oort[2], Richard Clough [3], Michael Seyani[3], Hesta McNeill[3], Mojgan Reza[3], Natalia Martinez-Soria[3], Salam A. Assi[4], Tatsiana V. Ramanouskaya[1], Constanze Bonifer [4] & Olaf Heidenreich [2,3,5✉]

The fusion oncogene *RUNX1/RUNX1T1* encodes an aberrant transcription factor, which plays a key role in the initiation and maintenance of acute myeloid leukemia. Here we show that the *RUNX1/RUNX1T1* oncogene is a regulator of alternative RNA splicing in leukemic cells. The comprehensive analysis of *RUNX1/RUNX1T1*-associated splicing events identifies two principal mechanisms that underlie the differential production of RNA isoforms: (i) RUNX1/RUNX1T1-mediated regulation of alternative transcription start site selection, and (ii) direct or indirect control of the expression of genes encoding splicing factors. The first mechanism leads to the expression of RNA isoforms with alternative structure of the 5'-UTR regions. The second mechanism generates alternative transcripts with new junctions between internal cassettes and constitutive exons. We also show that RUNX1/RUNX1T1-mediated differential splicing affects several functional groups of genes and produces proteins with unique conserved domain structures. In summary, this study reveals alternative splicing as an important component of transcriptome re-organization in leukemia by an aberrant transcriptional regulator.

[1] Department of Genetics, Faculty of Biology, Belarusian State University, 220030 Minsk, Republic of Belarus. [2] Princess Maxima Center for Pediatric Oncology, 3584 CS Utrecht, The Netherlands. [3] Wolfson Childhood Cancer Research Centre, Translational and Clinical Research Institute, Newcastle University, Newcastle upon Tyne NE1 7RU, UK. [4] Institute of Cancer and Genomic Sciences, College of Medical and Dental Sciences, University of Birmingham, Birmingham B15 2TT, UK. [5] Newcastle University Centre for Cancer, Newcastle University, Newcastle upon Tyne NE1 7RU, UK. ✉email: grinev_vv@bsu.by; o.t.heidenreich@prinsesmaximacentrum.nl

Alternative splicing increases the transcriptomic and proteomic complexity by generating distinct RNA isoforms with different roles and functions from a single gene[1]. When dysregulated, alternative splicing can contribute to tumorigenesis[2,3]. Dysregulation can take place on the level of alternative promoter choice leading the transcripts with altered 5'-termini, choice of alternative polyadenylation, and within transcript bodies by changes in splice site selection affecting exon composition or intron retention. Mis-splicing of genes has been found in multiple malignancies including carcinomas, neuroblastoma, chronic lymphoid leukemia and acute myeloid leukemia (AML)[4]. Mutations in spliceosome factor genes such as *SF3B1, SRSF2,* or *U2AF* are frequently found in myelodysplastic syndrome and are currently intensively examined for their therapeutic relevance[5–7].

Alternative splicing is also regulated by epigenetic marks[8,9]. Histone modifications and DNA methylation can affect exon usage by controlling the elongation speed of RNA polymerase II (RNA pol II) and, consequently, the choice of splice sites. Furthermore, they may affect the recruitment of splicing factors to chromatin through adapters such as CHD1. Moreover, chromatin modifications have been found to regulate the activity of alternative or cryptic transcriptional start sites (TSSs) in the genome[10]. Indeed, deregulation of alternative promoter usage has been recently identified as a common phenomenon in cancer[11]. However, underlying genetic reasons for alternative promoter choice are largely unknown.

The fusion oncogene *RUNX1/RUNX1T1* is expressed as a result of the chromosomal translocation t(8;21), which involves the *RUNX1* gene on chromosome 21 and the *RUNX1T1* gene on chromosome 8[12]. When expressed in haematopoietic cells, the fusion protein occupies more than 4000 genomic sites and forms transcription regulatory complexes by recruiting cofactors[13–18]. These complexes trigger a local remodeling of chromatin of a wide range of genes and thereby affect their expression[13,19,20]. In turn, the change of target gene expression leads to a block of cell differentiation, enhancement of self-renewal, modulation of the apoptosis and, eventually, to malignant transformation of t(8;21)-positive cells[15,21–23]. However, despite increasing knowledge of the molecular function of *RUNX1/RUNX1T1*, its impact on the leukemic transcriptome is still only incompletely understood. More generally, a potential role of leukemic fusion genes encoding epigenetic modulators such as RUNX1/RUNX1T1 has not yet been established for the regulation of alternative splicing.

To address this question, we performed perturbation experiments in t(8;21)-positive AML cells and examined the impact of *RUNX1/RUNX1T1* knockdown on the gene expression and RNA splicing at a global level. Our results demonstrate that *RUNX1/RUNX1T1* downregulation affects splicing of both direct target genes and in an indirect fashion. Mechanistically, changes in the production of RNA isoforms are implemented via (i) direct control of alternative transcription start site selection in target genes, and (ii) by direct or indirect control of the expression of the genes encoding splicing factors. Our modeling and experimental results indicate that oncoprotein-mediated differential splicing affects conserved domain structures in proteins, ultimately modulating leukemia-relevant processes such as nucleotide metabolism, cell adhesion and cell differentiation.

In conclusion, our results show that the fusion oncogene *RUNX1/RUNX1T1* controls alternative splicing in AML. These findings add to the complexity in the organization of leukemia-driving transcriptomes and set a paradigm for the role of fusion gene-encoded transcription factors in RNA processing.

## Results

**Change in the expression of the fusion oncogene *RUNX1/RUNX1T1* leads to differential splicing in the leukemic transcriptome**. To detect a potential association between RUNX1/RUNX1T1 and RNA splicing, we initially analyzed published RNA-seq, ChIP-seq and DNase-hypersensitive-site-seq (DHS-seq) data sets for links between RUNX1/RUNX1T1 and RNA processing (Supplementary Data 1). Functional annotation by DAVID uncovered a highly significant enrichment of RUNX1/RUNX1T1 binding at gene loci that are subject to alternative splicing and splice variants (false discovery rate adjusted *p*-value or *q*-value, $4.0 \times 10^{-23}$ and $1.5 \times 10^{-12}$, respectively; Supplementary Fig. 1a, Supplementary Data 2)[24]. These findings are in line with the observation that RUNX1/RUNX1T1 may interact with several splicing factors implying a direct link between RUNX1/RUNX1T1-regulated transcription and RNA processing[15]. Furthermore, we observed that RUNX1/RUNX1T1 binding sites are often present within genes encoding splicing-associated factors or in their immediate vicinity (Supplementary Data 3, 4). The list of such genes includes both classical splicing regulatory genes such as *HNRNPM, RBFOX2, SF3A3, SRSF5,* and genes encoding the components of nuclear speckles including *RBM6, SNRPD3, SNU13,* which are considered the area of intensive splicing[25,26]. Moreover, gene set enrichment analysis (GSEA) suggests that knockdown of *RUNX1/RUNX1T1* affects RNA binding proteins and snRNP assembly, and is associated with impaired mRNA processing and, in particular, splicing pathways (Fig. 1a; Supplementary Fig. 1b)[27]. These findings strongly suggest a regulatory role of this leukemic fusion protein in mRNA splicing and predict changes in splicing pattern in response to perturbing RUNX1/RUNX1T1 activity.

To identify differential splicing events regulated by RUNX1/RUNX1T1, we analyzed RNA-Seq data of the t(8;21)-positive AML cell line Kasumi-1 following *RUNX1/RUNX1T1* knockdown (Supplementary Data 1, GSE54478). Knockdown was achieved by electroporation with the short interfering RNA (siRNA) siRR targeting the fusion site of the *RUNX1/RUNX1T1* mRNA, while a mismatch siRNA, siMM, served as a control[28] (Fig. 1b).

The differential splicing events were detected at the level of both exons and exon-exon junctions (EEJs). Differentially used exons (diffUEs) were identified by use of DEXSeq, limma/diffSplice and JunctionSeq, while differentially used exon-exon junctions (diffEEJs) were identified using limma/diffSplice and JunctionSeq functionality[29–31]. This combined approach detected a comprehensive set of differential splicing events in dependence on the RUNX1/RUNX1T1 status.

Using our pipeline, we developed a list of non-overlapping genomic fragments by analyzing 511,363 non-overlapping bins of human canonical exons annotated in Ensembl[32]. This exonic list was extended by 3540 non-overlapping bins of retained introns detected in 2401 genes of the Kasumi-1 cells, of which 25% were differentially retained between *RUNX1/RUNX1T1* knockdown and control cells (Supplementary Data 5). Of note, knockdown affected intron retention in both ways: 419 introns were exclusively detected in RUNX1/RUNX1T1-depleted cells, while 469 introns were only retained in control cells (Fig. 1c, d; Supplementary Fig. 1c, d). These condition-specific retained introns show a small but noticeable difference in expression (Supplementary Fig. 1e–i).

In addition to differentially retained introns, we identified 395 differentially used exons (diffUEs) in the Kasumi-1 transcriptome following *RUNX1/RUNX1T1* knockdown (Fig. 1e, f; Supplementary Fig. 1j; Supplementary Data 6). The list of the identified diffUEs included 387 canonical exons and 8 retained introns affecting 275 genes (Fig. 1g, h). Interestingly, while these diffUEs were uniformly distributed across non-coding RNAs, in protein-

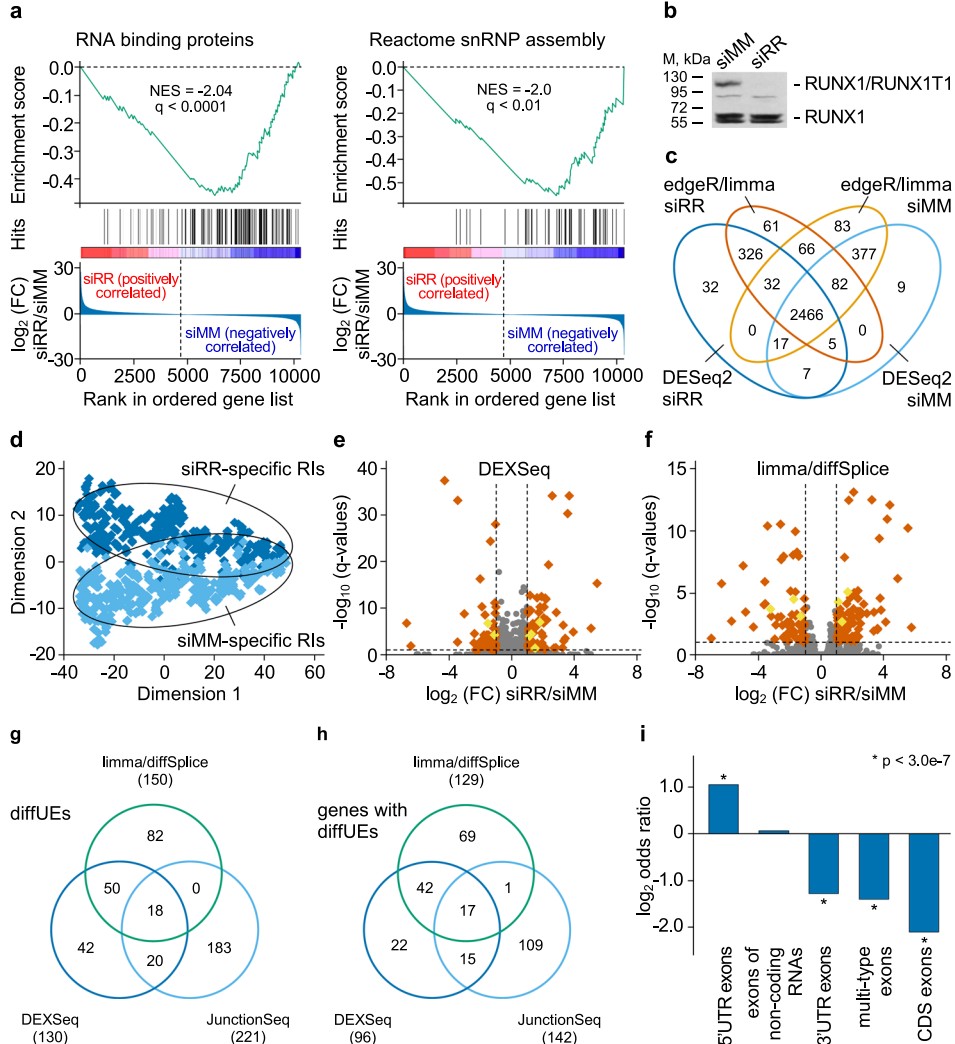

**Fig. 1 Knockdown of *RUNX1/RUNX1T1* affects exon usage in Kasumi-1 cells. a** Gene set enrichment analysis plots for RNA binding proteins[55] and the Reactome snRNP assembly pathway. NES, normalized enrichment score; q, false discovery rate. **b** Western blot of RUNX1 and RUNX1/RUNX1T1 protein levels in Kasumi-1 cells following RUNX1/RUNX1T1 knockdown. A representative result from one of the three experiments is shown. siMM, mismatch siRNA; siRR, RUNX1/RUNX1T1 siRNA. **c** Distribution of retained introns (RIs) between the transcriptomes of the siRR-treated and siMM-treated Kasumi-1 cells. The RIs were detected using the DESeq2 and edgeR/limma algorithms. **d** Scatter plot showing the distribution of siRR-specific and siMM-specific RIs after multidimensional reduction of RIs expression data using t-distributed stochastic neighbor embedding. The RIs were detected using the edgeR/limma algorithm, and very similar results were observed with DESeq2. **e, f** Vulcano plots showing differential usage of exons identified by DEXSeq **e** or limma/diffSplice **f** algorithms. Orange and yellow rhombi indicated differentially used canonical exons or RIs, respectively, with more than 2-fold change and q < 0.1. **g** Venn diagram showing overlap between differentially used exons (diffUEs) identified by DEXSeq, limma/diffSplice and JunctionSeq algorithms. **h** Venn diagram showing overlap of genes with diffUEs identified by the DEXSeq, limma/diffSplice and JunctionSeq algorithms. **i** Classification of diffUEs according to the functional type of exons. Genomic coordinates of the reference exons were extracted from Ensembl models of the human genes. diffUEs were intersected with reference exons and the overlaps counted. Fisher's two-sided exact test was performed to detect over-represented or under-represented exon types among the diffUEs against non-differential exons (non-diffUEs).

coding transcripts they were significantly enriched for 5'-UTR exons and depleted for CDS, 3'-UTR and multi-type exons (Fig. 1i).

At the level of EEJs, we detected 177 differentially used EEJs (diffEEJs) upon *RUNX1/RUNX1T1* knockdown that were distributed over 150 genes (Fig. 2a–c; Supplementary Fig. 2a; Supplementary Data 7). These diffEEJs belonged to the main modes of alternative splicing (Fig. 2d) and associated with all functional types of exons (Fig. 2e). In contrast to genes with condition-specific retained introns, only 55 of 378 genes with diffUEs or diffEEJs showed changes in total transcript levels upon *RUNX1/RUNX1T1* knockdown (Supplementary Fig. 2b, c).

Notably, use of two different algorithms limma/diffSplice and JunctionSeq for the analysis of diffEEJs produced only minimally overlapping results (Fig. 2b, c). This is due to the differences in the pre-processing, filtering, normalization and transformation of the data, as well as in the use of different statistical models at the step of identification of diffEEJs by the algorithms. However, diffEEJs identified by both limma/diffSplice and JunctionSeq could be confirmed using qRT-PCR (Supplementary Data 8).

**RUNX1/RUNX1T-associated alternatively spliced variants are distinctive features in primary leukemia samples.** To examine whether diffEEJs detected in Kasumi-1 cells are also present in

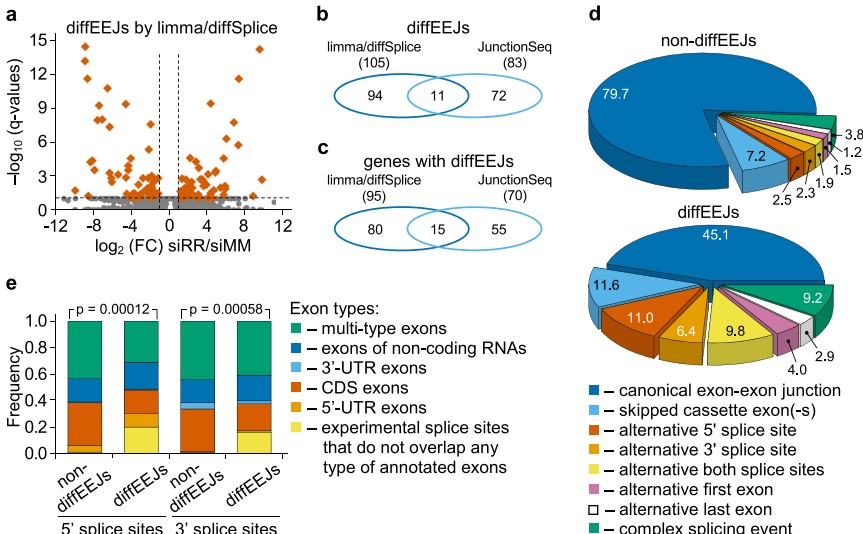

**Fig. 2 Knockdown of *RUNX1/RUNX1T1* causes changes in exon-exon linkages in leukemic cells. a** Vulcano plot showing differential splicing of exons identified by limma/diffSplice algorithm. Orange rhombi indicated differentially used exon-exon junctions (diffEEJs) with more than 2-fold change and *q* < 0.1. **b** Venn diagramme showing the overlap between diffEEJs identified by limma/diffSplice and JunctionSeq algorithms. **c** Venn diagramme showing overlap of genes with diffUEs identified by limma/diffSplice and JunctionSeq algorithms. **d** Pie charts showing the classification of diffEEJs and non-differential exon-exon junctions (non-diffEEJs) according to the modes of alternative splicing. Numbers indicate the percentage of splicing events assigned to a particular mode of splicing. Complex splicing means several (two or more) alternative splicing events occurred simultaneously. **e** Distribution of the splice sites with diffEEJs and non-diffEEJs among the various functional types of exons. P-values were calculated with $\chi^2$ test. Panels **a** to **e** are based on data from Kasumi-1 cells.

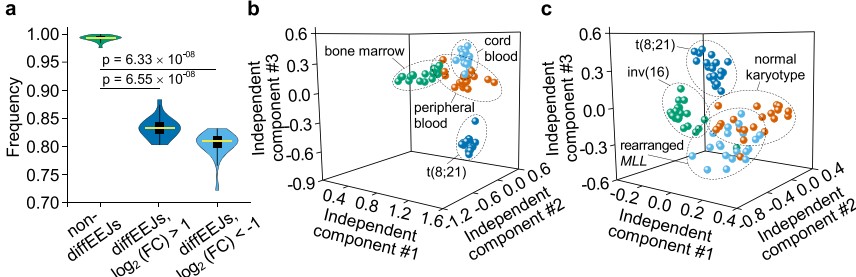

**Fig. 3 RUNX1/RUNX1T1-associated alternatively spliced variants are distinctive features in primary leukemia samples. a** Violin plot showing the enrichment level of Kasumi-1 exon-exon junctions in the transcriptome of primary t(8;21)-positive leukemia blasts from 20 patient samples. *P*-values were calculated with two-sided Mann–Whitney U test. **b, c** Independent component analysis of t(8;21)-positive AML cells with normal CD34-positive cells **b** or other AML subtypes **c**. This analysis included 20 samples of each cell type. For each sample, the expression of 84,743 exon-exon junctions was quantified.

primary leukemia blasts, we selected publicly available RNA-Seq samples from 20 patients with t(8;21)-positive AML (Supplementary Data 1, GSE62190 and GSE67040). With this dataset we detected about 80% of Kasumi-1-associated diffEEJs that are abundant in the transcriptome of primary t(8;21)-positive leukemia blasts (Fig. 3a). This extensive overlap is in line with the previously observed high correlation between t(8;21)-positive cell lines and patient material regarding gene expression and chromatin accessibility, and proves Kasumi-1 as a relevant model system for RUNX1/RUNX1T1-regulated gene expression[19,21,23,33].

Next, we tested if these splicing events were specific features of t(8;21)-positive AML by comparing splicing events found in t(8;21) AML cells with those in normal cells or cells from other AML subtypes (Supplementary Data 1). To that end we analyzed RNA-Seq data from normal CD34-positive cells obtained from human bone marrow (GSE102881, GSE111085, GSE63569, and GSE69239), umbilical cord blood (GSE48846, GSE69905, and GSE71689) and peripheral blood (GSE102881, GSE107218, GSE68925, GSE87285, GSE90552 and GSE92274) in addition to AML cells with normal karyotype (GSE49642 and GSE52656),

inv(16) (GSE108266, GSE62190, and GSE67039) or a rearranged *MLL* locus (GSE52656, GSE62190, and GSE67039). With this comprehensive dataset, independent component analysis revealed three independent subsets of EEJs that clearly separate t(8;21)-positive AML cells from normal CD34-positive cells (Fig. 3b; Supplementary Data 9). We found that 53% of diffEEJs identified in Kasumi-1 were present in these independent components with high statistical significance (odds ratio 5.42, $p < 2.2 \times 10^{-16}$). A similar result was obtained when comparing t(8;21)-positive cells with other types of leukemia (Fig. 3c; Supplementary Data 10) with 48% of those diffEEJs identified in Kasumi-1 contributing to the discrimination of t(8;21)-positive AML from other AML subtypes (odds ratio 2.67, $p = 4.1 \times 10^{-12}$).

Together, these results clearly indicate that RUNX1/RUNX1T1-dependent splicing events are characterizing features of the transcriptome of t(8;21)-positive AML cells. The occurrence of these events in combination with selected RUNX1/RUNX1T1-independent splicing events distinguishes t(8;21)-positive from t(8;21)-negative AML cells and normal haematopoietic stem and progenitor cells.

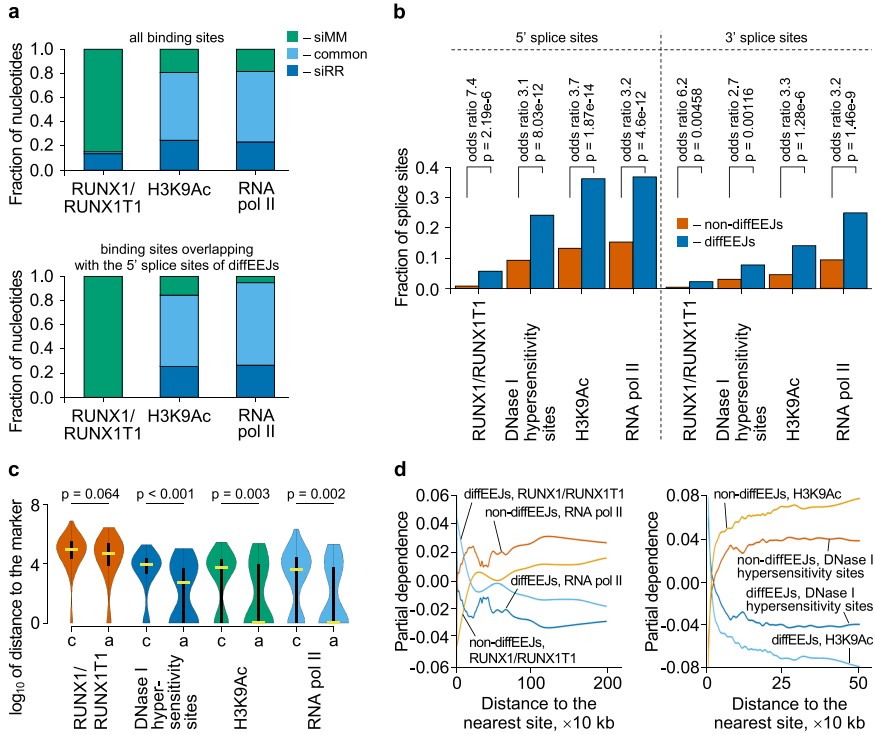

**Fig. 4 *RUNX1/RUNX1T1*-dependent differential splicing events in Kasumi-1 cells are associated with open chromatin. a** Column graph showing the distribution of RUNX1/RUNX1T1, H3K9Ac and RNA pol II occupied sites with or without RUNX1/RUNX1T1 knockdown. The top panel shows the distribution for all sites, the bottom panel only shows distribution of binding sites overlapping with the 5' splice sites of diffEEJs. **b** Association of diffEEJs splice sites with open chromatin. *P*-values were calculated with Fisher's two-sided exact test. **c** Violin graph depicting the positional relationships between canonical (c; *n* = 117) and alternative (a; *n* = 57) 5' splice sites of diffEEJs and marks of open chromatin. *P*-values were calculated with two-sided Mann–Whitney U test. Yellow horizontal thick lines represent the median of distances distribution, the black vertical rectangular boxes show the interquartile range, and the black vertical lines are the 95% confidential interval. **d** Effect of the proximity to RUNX1/RUNX1T1 binding sites, RNA pol II peaks, H3K9Ac histone modifications and DNase I hypersensitivity sites on the probability of the diffEEJs and non-diffEEJs in the transcriptome of leukemia cells.

**Modulation of local epigenetic marks by RUNX1/RUNX1T1 influences alternative splicing**. Next, we determined the mechanisms through which RUNX1/RUNX1T1 controls splicing events in t(8;21)-driven leukemia. Binding of RUNX1/RUNX1T1 at specific genomic sites affects the local chromatin status, including changes in histone modifications and chromatin accessibility[14,19,20] (Fig. 4a, H3K9Ac data). These epigenetic changes were associated with the redistribution of RNA pol II peaks indicating changes in transcription rates upon RUNX1/RUNX1T1 knockdown (Fig. 4a, RNA pol II data).

Since transcription kinetics can affect alternative splicing[34,35], we examined the potential impact of RUNX1/RUNX1T1-dependent chromatin organization and transcription upon splicing events in more detail. Splice sites of diffEEJs were significantly enriched in the regions of open chromatin (marked by DNase I hypersensitivity sites and sites of H3K9Ac histone modification) and decelerated RNA pol II elongation compared to the splice sites of non-diffEEJs (Fig. 4b). Comparison of canonical splice sites with diffEEJs demonstrated a closer proximity of alternative 5' splice sites (but not 3' splice sites) to open chromatin regions, and sites bound by H3K9Ac and RNA pol II (Fig. 4c). Similarly, alternative 5' splice sites showed also a trend towards closer proximity to RUNX1/RUNX1T1 sites. These data suggest that splice sites of diffEEJs, which are occupied by H3K9Ac and RNA pol II, are controlled by RUNX1/RUNX1T1.

All of the above findings indicate a close association between chromatin state and differential splicing following *RUNX1/RUNX1T1* knockdown. Since the epigenetic landscape affects alternative splicing only in cooperation with sequence features of

pre-mRNA[36], we integrated epigenetic data with multivariate sequence data and calculated the partial dependence for each epigenetic mark using a random forest approach (see Supplementary Methods for further details). This analysis demonstrates that differential splicing is promoted by proximity to RUNX1/RUNX1T1, RNA pol II, and H3K9Ac sites, as well as DNase I hypersensitivity sites (Fig. 4d) and highlights the influence of the chromatin state on differential splicing following RUNX1/RUNX1T1 knockdown.

**Delayed splicing is sensitive to regulation by RUNX1/RUNX1T1**. The majority of splicing events are co-transcriptional. However, in some cases of regulated alternative splicing, intron removal can be delayed until the release of pre-mRNA from the site of transcription[37,38]. With this in mind, we analyzed nascent RNA in Kasumi-1 cells for EEJs that were shared between nascent and total, i.e., mature RNA (Fig. 5a). These EEJs were grouped into two classes. The first class (non-diffEEJs) included the nascent EEJs which did not become differential EEJs at the level of total RNA. In contrast, the second class (diffEEJs) collected events that were identified as diffEEJs at the level of total RNA.

Interestingly, expression of diffEEJs (affected by *RUNX1/RUNX1T1* knockdown; Fig. 5b) showed a much stronger bimodal distribution resulting in a lower median compared to non-diffEEJs (Fig. 5c, Supplementary Table 1). For the nascent RNA, the number of low abundant diffEEJs substantially exceeded that of high abundant ones, while the opposite was true for total RNA. In contrast, the distribution pattern of non-diffEEJs shifted only slightly towards higher abundances. Importantly, neither

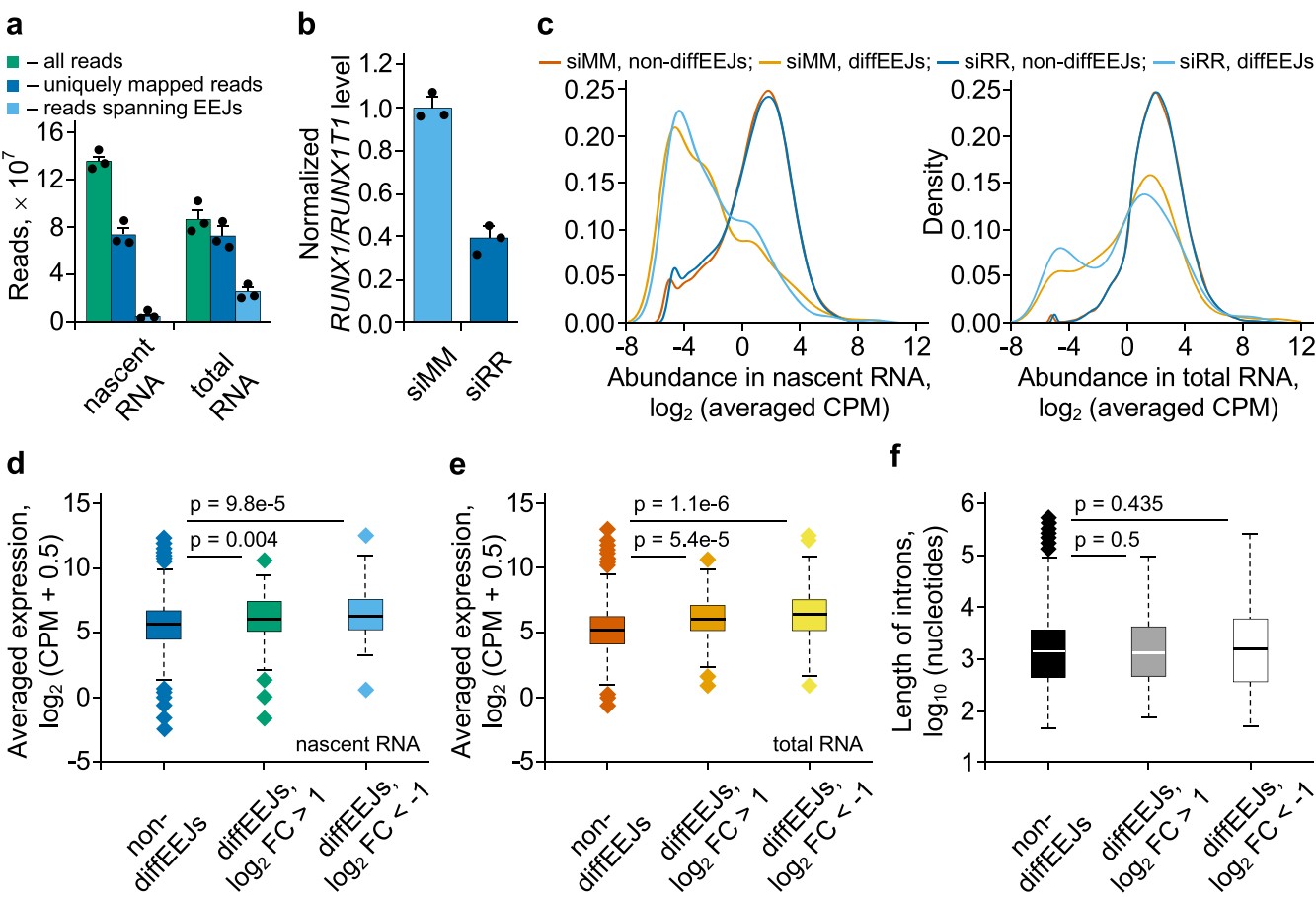

**Fig. 5 Formation of diffEEJs in Kasumi-1 cells is delayed compared to non-diffEEJs. a** The basic alignment statistics for the nascent RNA and total RNA datasets obtained from the siMM-treated leukemia cells. Bars represent mean ± SD of three independent experiments. Very similar results were observed with siRR-treated cells. **b** siRNA-mediated knockdown of *RUNX1/RUNX1T1* as measured by nascent RNA expression. Bars represent mean ± SD of three independent experiments. **c** Multidensity plots of the abundance of EEJs identified in the nascent RNA (left panel) or total RNA (right panel). These plots are based on the RNA-Seq data obtained from the siMM-treated or siRR-treated leukemia cells. All EEJs identified at the level of nascent RNA and total RNA were grouped (see main text for further explanation of classification procedure) into non-diffEEJs or diffEEJs, and the abundance of these events was plotted. Each line represents the data averaged over the three independent biological replicas. **d** Expression of genes without (non-diffEEJs genes, $n = 10851$) or with differential splicing (upregulated or downregulated diffEEJs genes, $n = 85$) at nascent RNA level. **e** Expression of genes without (non-diffEEJs genes, $n = 9836$) or with differential splicing (upregulated or downregulated diffEEJs genes, $n = 150$) at total RNA level. **f** Comparison of intron lengths between non-diffEEJs ($n = 104925$) and diffEEJs ($n = 177$). In **d**, **e**, and **f**, boxplots summarize the expression data averaged over the three independent biological replicas. In each boxplot, horizontal line represents the median of expression distribution, box shows the interquartile range, and whiskers are the minimum and maximum. *P*-values were calculated with two-sided Mann–Whitney U test.

expression levels of differentially spliced versus non-differentially spliced genes nor the distribution of intron lengths differed substantially between non-diffEEJs and diffEEJs and, thus, are not the cause for the observed shifts (Fig. 5e, f). Therefore, these findings suggest the presence of differential splicing events which undergo delayed processing. Furthermore, they imply that delayed splicing is more sensitive to genomic and transcriptomic perturbations induced by the changes in the *RUNX1/RUNX1T1* expression.

**Differential splicing produces proteins with unique conserved domain structures and affects multiple cellular processes**. Next, we assessed in silico the potential functional consequences of RUNX1/RUNX1T1-associated differential splicing. Enrichment analysis with Gene Ontology gene sets suggested that differential splicing particularly affects genes controlling synthesis, transport, targeting and localization of proteins, cell adhesion and granulocyte activation (Fig. 6a).

To further detail above functional findings, we used RNA-Seq data (Supplementary Data 1, GSE54478) and Cufflinks/

Cuffdiff algorithm to assemble of full-length RNA molecules, as well as to quantify their expression in Kasumi-1 cells. We found that genes affected by RUNX1/RUNX1T1-dependent differential splicing encode 752 transcripts, of which 214 isoforms were differentially spliced. Here and below, the term "differentially spliced isoform" indicates a molecule of RNA with at least one differential exon-exon junction (upregulated or downregulated), while a non-differential isoform is a transcript lacking any diffEEJs.

Of all above transcripts, 75% were potentially protein coding, while the rest were non-coding RNAs (Fig. 6b, c, upper panel). We aligned in silico translated proteins against the human NCBI RefSeq proteins (release 92, last modified on 18 March 2019) and the newest non redundant releases of GenBank CDS translations, UniProtKB/SwissProt, Protein Data Bank, Protein Information Resource and Peptide/Protein Sequence Database proteins (updated on 11 April 2019) using NCBI blastp[39]. Out of 1055 in silico translated proteins, 722 proteins were 90% or more identical to the proteins annotated in the public databases (Fig. 6c, bottom panel). Such proteins form a clear peak on the

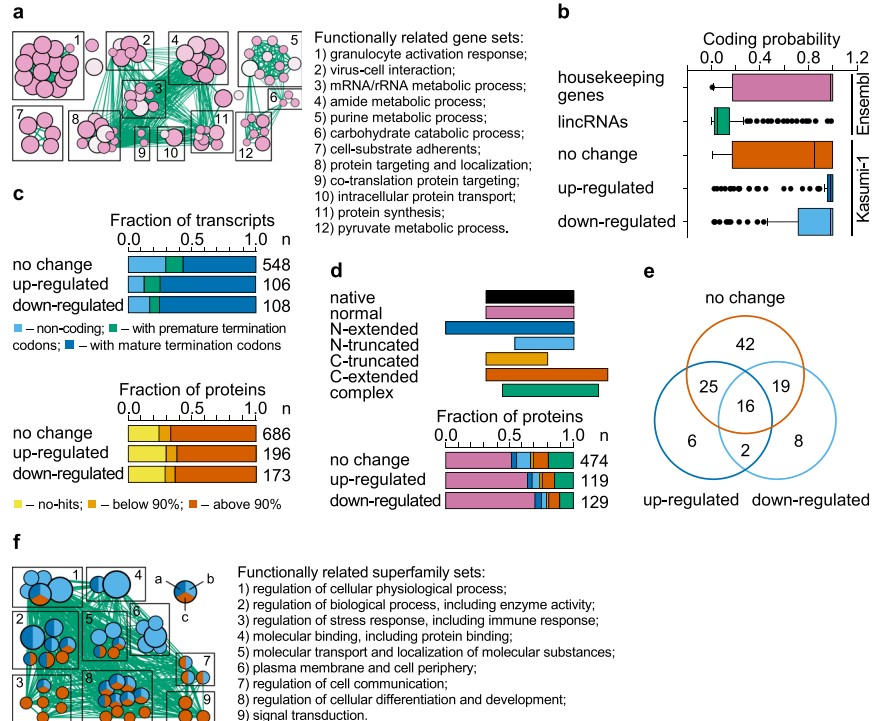

**Fig. 6 Fusion protein-controlled differential splicing expands the protein space. a** Enrichment map of the Gene Ontology-enriched gene sets across all the genes with differential expression and/or differential splicing. Nodes represent significantly enriched gene sets and node size is proportional to the number of members in a gene set. Edges indicate the gene overlap between the nodes, and the thickness of the edges is equivalent to the degree of the gene overlap between the nodes. Functionally related gene sets are clustered and named. **b** Box plot depicting the coding potential of 500 randomly selected transcripts each from housekeeping genes and lincRNA genes, and all 752 transcripts of genes affected by differential splicing in Kasumi-1 cells. In each boxplot, horizontal line represents the median of probability distribution, box shows the interquartile range, and whiskers are the minimum and maximum. **c** Column graphs depicting the classification of coding sequences in transcripts (upper panel) and of in silico translated proteins (lower panel) affected by differential splicing. In the upper panel, all transcripts were classified on non-coding (no significant open reading frames were found), protein-coding with premature termination codons and protein-coding with mature termination codons. In the bottom panel, all protein-coding transcripts were in silico translated and predicted proteins were blastp aligned against non-redundant set of human proteins. Protein identities are indicated at the bottom. **d** Column graphs showing the impact of RUNX1/RUNX1T1 on the frequency of protein variants. **e** Venn diagramme visualizing the distribution of the 118 domain superfamilies among the in silico predicted proteins. **f** Enrichment map of the domain-centered Gene Ontology-enriched superfamily sets across all the genes with differential splicing. Nodes represent significantly enriched superfamily sets and node size is proportional to the number of members in a superfamily set. Edges indicate the superfamily overlap between the nodes, and the thickness of the edges is equivalent to the degree of the superfamily overlap between the nodes. Functionally related superfamily sets are clustered and named. Pie chart coloring: (a) upregulated, (b) downregulated, and (c) no change.

curve of identity distribution and, to reduce noise, only these proteins were selected for downstream analysis.

A substantial fraction of in silico translated proteins was similar to native human proteins (Fig. 6d). At the same time, there were also different extended and truncated isoforms of proteins. Such isoforms can potentially include new domains or may lack domains important for proper protein binding, localization and function, with other functional regions remaining unaffected.

To further structural detail, we assigned SCOP domains for highly identical in silico translated proteins using the SUPER-FAMILY hidden Markov models, and we identified 225 conserved domains within these proteins (Structural Classification of Proteins database, release 1.75, June 2009)[40,41]. Identified domains belong to 146 domain families. About 45% of these domain families are common for proteins translated from differentially (upregulated or downregulated) and non-differentially (no change) spliced RNA isoforms. However, 16% of domain families are associated with differentially spliced RNA isoforms. This uniqueness of differential proteins is robust and is retained even when domains are combined into domain super-families (Fig. 6e).

Finally, we used a domain-centric Gene Ontology method for functional annotation of the domain architectures of in silico predicted proteins[42]. We applied the enrichment test separately for proteins translated from differentially spliced (upregulated or downregulated) or non-differentially spliced RNA isoforms, and we integrated the results using EnrichmentMap[43]. This in silico approach showed that the domains encoded by differential RNA isoforms are mainly involved in the global regulation of cellular processes, molecular binding (including protein binding), transport and localization of molecular substances, and cell communications (Fig. 6f).

**RUNX1/RUNX1T1 affects differential splicing by controlling alternative promoter usage.** RUNX1/RUNX1T1 binding sites are enriched in promoter and intronic regions indicating that the fusion protein directly affects transcription of multiple genes[14,19,21]. Consistently, in our dataset of diffUEs and diffEEJs, we found that 38% of genes with differential splicing were also enriched for fusion protein binding sites (Supplementary Data 6, 7). This enrichment was significantly higher than that for non-differentially spliced genes (odds ratio 2.47 and $p = 2.5 \times 10^{-15}$ according to two-sided Fisher's exact test). Based on these data,

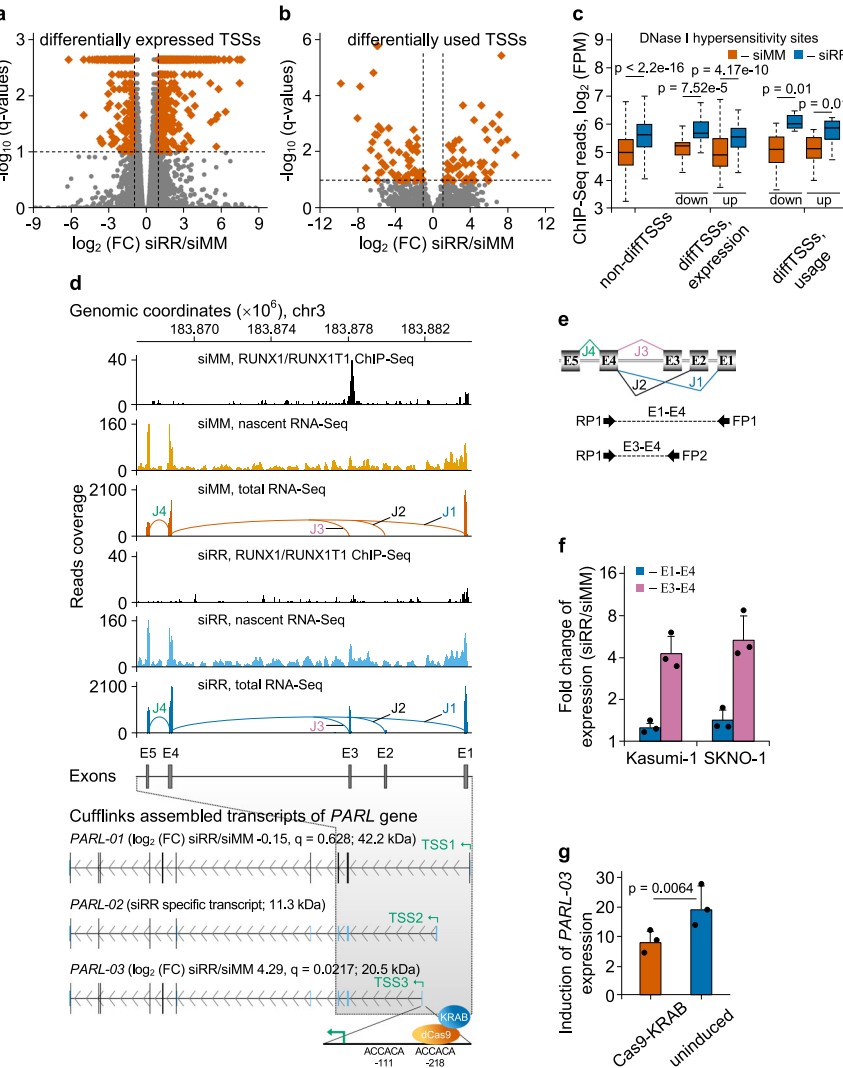

**Fig. 7 RUNX1/RUNX1T1 affects differential splicing by controlling alternative transcription start sites (TSSs). a** Vulcano plot of the differential expression from various TSSs in the genome of Kasumi-1 cells following *RUNX1/RUNX1T1* knockdown. Differentially expressed TSSs (at least 2-fold change in expression, $p < 0.005$, $q < 0.1$) are shown by vermilion squares. **b** Differential usage of alternative TSSs in the genome of Kasumi-1 cells after *RUNX1/RUNX1T1* knockdown. Differentially used TSSs (at least 2-fold change in expression, $p < 0.0007$, $q < 0.1$) are shown by vermilion squares. **c** Enrichment of DNase I hypersensitivity site-Seq reads in the genomic regions surrounding the non-differential TSSs, differentially expressed TSSs or differentially used TSSs. In this analysis, only TSSs genomic regions that overlap the RUNX1/RUNX1T1 binding peaks were used. In each boxplot, horizontal line represents the median of expression distribution, box shows the interquartile range, and whiskers are the minimum and maximum. *P*-values were calculated with two-sided Mann–Whitney U test. **d** IGV screen shot showing the RUNX1/RUNX1T1 binding and RNA reads from nascent and total RNA-Seq at the 5′-terminal region of PARL locus for siMM-treated and siRR-treated Kasumi-1 cells. Exons and exon-exon junctions are designated by the letters E and J, respectively, and numbered. Differentially used junction J3 is highlighted in purple. Bottom scheme depicting Cufflinks assembled representative full-length transcripts of *PARL* gene expressed in Kasumi-1 cells. For each transcript, Cuffdiff based log$_2$ (FC) siRR/siMM and *q*-values, as well as the size of in silico predicted proteins are indicated in parentheses. In addition, the positions of the Cuffdiff determined transcription start sites are also shown. At the bottom of panel, the scheme showing target location of dCas9-KRAB which was targeted to the second RUNX1 consensus site. **e** Schematic representation of primer binding positions for qPCR-based validation of the differential splicing in *PARL* gene. The structure of expected amplicons is shown at the bottom of the figure. **f** qPCR-based validation of the differential splicing of PARL transcripts under the two siRNA treatment conditions in the transcriptome of two t(8;21)-positive cell lines. Bars represent mean ± SD of three independent experiments. **g** Induction of *PARL-03* transcription after RUNX1/RUNX1T1 knockdown with or without induction of dCas9-KRAB. Bars represent mean ± SD of three independent experiments. *P*-value was calculated with two-sided Student's t test.

we hypothesized that differential splicing events can arise from the selection of alternative TSSs under the control of the fusion protein. Indeed, we identified 775 TSS associated with differential transcription (more than 2-fold change, q < 0.1) in the genome of Kasumi-1 cells following *RUNX1/RUNX1T1* knockdown (Fig. 7a). Among those, knockdown of the *RUNX1/RUNX1T1* led to differential usage of 112 alternative TSSs (more than 2-fold change, *q* < 0.1) (Fig. 7b). Loss of RUNX1/RUNX1T1 increased chromatin accessibility and RUNX1 occupation at alternative

TSSs to a similar extend as in the total group of TSSs driving differential transcription (Fig. 7a, Supplementary Fig. 3a). Thus, perturbation of RUNX1/RUNX1T1 reorganizes the chromatin structure of alternative TSSs leading to their transcriptional activation.

The cohort of genes where RUNX1/RUNX1T1 represses alternative TSSs includes the NAD kinase gene *NADK*, the *RUNX1T1* paralogue *CBFA2T3*, the kinase gene *RPS6KA1 and* the rhomboid kinase gene *PARL* (Supplementary Data 6, 7). In

each case, loss of RUNX1/RUNX1T1 activated an alternative TSS that either exclusively affected the 5'-UTR as in the case of *NADK*, was predicted to change only the N-terminal amino acids of the corresponding protein (e.g., *CBFA2T3, RPS6KA1*) or may have caused larger N-terminal deletions (e.g., *PARL*).

For instance, RUNX1/RUNX1T1 occupies the *PARL* gene that encodes a kinase controlling mitophagy and apoptosis and that is associated with clinical outcome of AML (Supplementary Fig. 3b)[44,45]. The binding site of RUNX1/RUNX1T1 locates in the first intron 6.6 kb downstream of the canonical *PARL* TSS1. Knockdown of RUNX1/RUNX1T1 eliminated its binding to this genomic element and activated 3'-proximally located TSS2 and TSS3 (Fig. 7d, e). In particular transcription of a non-canonical exon from TSS3 and the corresponding exon-exon junction J3 yielded alternative transcript *PARL-03* (Fig. 7d). The observed increase in total *PARL* transcript and protein levels in t (8;21) blasts upon RUNX1/RUNX1T1 knockdown was mainly due to alternative transcript *PARL-03* (Supplementary Fig. 3c, d). Furthermore, junction J3 was a specific feature of the transcriptomes of primary AML subtypes with a more mature immunophenotype than t(8;21) AML, but of neither t(8;21)-positive AMLs nor normal CD34-positive haematopoietic stem and progenitor cells (Supplementary Fig. 3e). These data support the notion that the non-canonical TSS3 is activated in some AMLs, but is repressed in the t(8;21) AML subtype.

We further examined the functionality of this element by targeting an endonuclease-deficient dCas9-KRAB repressor to a site 200 bp upstream of TSS3 (Fig. 7d). Expression of dCas9-KRAB was induced by addition of doxycycline to the medium. When compared to mismatch siRNA-treated Kasumi-1 cells, knockdown of RUNX1/RUNX1T1 caused twentyfold increased expression of *PARL-03* without significantly affecting canonical *PARL-01* levels (Fig. 7g). Induction of dCas9-KRAB did not affect *PARL-01* expression with 0.98 and 1.01fold changes for siMM and siRR-treated cells, respectively. However, dCas9-KRAB induction significantly reduced the induction of *PARL-03* transcription that was caused by RUNX1/RUNX1T1 loss (Fig. 7g). These findings confirm that transcription of *PARL-03* is regulated by a promoter located 6.6 kb in canonical intron 1 and further supports the notion that binding of RUNX1/RUNX1T1 to this element represses its transcription-activating potential.

Alternatively, loss of RUNX1/RUNX1/T1 binding can also result in a repression instead of an activation of alternative TSSs as in the case of *LIMS1*, where repression of TSS2 led to a substantially decreased level of LIMS1 protein isoform e (Supplementary Fig. 4). Thus, similar to canonical regulation of transcription, RUNX1/RUNX1T1 can both repress and activate alternative TSSs.

**Activation of an alternative TSS in the *RPS6KA1* gene compensates for RUNX1/RUNX1T1 loss.** Another example for RUNX1/RUNX1T1-exerted control of an alternative TSS is provided by the *RPS6KA1* locus. RUNX1/RUNX1T1 binds to a site in intron 1 located 12 kb downstream of TSS1 (Fig. 8a). Loss of RUNX1/RUNX1T1 binding results in a threefold increase in transcript and protein levels in t(8;21) AML cells (Fig. 8a, Cufflinks panel; Fig. 8b, c). This increase is mainly due to the twentyfold activation of TSS2 located in direct proximity downstream of the RUNX1/RUNX1T1 binding site.

*RPS6KA1* encodes ribosomal kinase 1 (RSK1, pp90[RSK1]), which promotes cell proliferation and survival by linking MAPK signaling to the MTOR pathway and inhibiting pro-apoptotic and antiproliferative factors including BAD and CDKN1B (also known as p27)[46]. Interestingly, RUNX1/RUNX1T-positive AML expresses significantly lower amounts of *RPS6KA1* transcripts

than other AML subtypes (Fig. 8d). Furthermore, high expression levels of this gene correlate with a worse clinical outcome (Fig. 8e). We, therefore, wondered if the upregulation of RPS6KA1 may rescue t(8;21) AML cells from transient RUNX1/RUNX1T1 loss and the therewith connected to impaired self-renewal capacity.

To that end we took advantage of the RSK-specific inhibitor BI-D1870 and examined the impact of RPS6KA1 inhibition on the clonogenic potential of t(8;21) AML cells as a surrogate read-out for leukemic self-renewal. In general, RPS6KA1 inhibition substantially impaired clonogenic growth in a dose-dependent fashion (Fig. 8f). However, RUNX1/RUNX1T1-knockdown sensitizes leukemic cells to RPS6KA1 inhibition: BI-D1870 alone reduced clonogenicity with an apparent IC$_{50}$ of 600 nM, a preceding knockdown of RUNX1/RUNX1T1 lowered this value to 100 nM. In conclusion, alternate TSS usage and therewith linked increased expression of RPS6KA1 compensated for the loss of RUNX1/RUNX1T1 strongly suggesting RPS6KA1 as a pharmacologic target for t(8;21) AML. As target identification usually focuses on genes and proteins with elevated expression levels[23], these data also prove the value of considering gene products repressed by an oncogene as candidates for therapeutic intervention.

**RUNX1/RUNX1T1 controls alternative splicing through differential expression of splicing factors.** Around 60% of the differential splicing events in Kasumi-1 cells cannot be explained by proximal RUNX1/RUNX1T1 binding indicating that they are not under direct control of the fusion protein. However, we and others have previously shown that t(8;21)-positive leukemia cells have a specific signature of genes encoding splicing factors and mRNA surveillance genes expression that may affect splicing patterns and abundance of RNA isoforms[35,47–52]. This signature is so unique that it allows distinguishing leukemic from normal hematopoietic cells (Supplementary Fig. 5a, b). Moreover, correlation analysis confirmed highly coordinated expression of genes encoding splicing factors and mRNA surveillance genes with *RUNX1/RUNX1T1* in t(8;21)-harboring AML cells (Supplementary Fig. 5c). Furthermore, RUNX1/RUNX1T1 binding sites were often present within the genes encoding splicing-associated factors or in their immediate vicinity (Supplementary Data 3, 11). Therefore, we speculated that RUNX1/RUNX1T1-dependent changes in expression of RNA processing genes may be responsible for differential splicing events in Kasumi-1 cells.

By combining of RNA-Seq (Supplementary Data 1, GSE54478) and microarray (Supplementary Data 1, GSE29223) data and three algorithms (Cufflinks/Cuffdiff, DESeq2 and edgeR/limma), we identified 42 genes encoding splicing factors and 14 mRNA surveillance genes with statistically significant differential expression in Kasumi-1 cells following the *RUNX1/RUNX1T1* knockdown (Supplementary Data 11; Fig. 9a, b). Of those, 16 genes including *USB1* are associated with RUNX1/RUNX1T1 binding sites implying them as direct target genes (Fig. 9c, Supplementary Data 11).

However, genes encoding splicing factors and mRNA surveillance genes may also indirectly be regulated via fusion protein-mediated control of transcription factors expression. To test this hypothesis, we reconstructed gene regulatory network for those genes encoding splicing factors and mRNA surveillance genes with significantly changed expression upon RUNX1/RUNX1T1 depletion (Fig. 9d). This network was inferred from microarray, RNA-Seq and bioinformatics data (see Supplementary Methods for further details). The reconstructed network suggests a complex expression interplay between the fusion protein, transcription factors and the factors involved in splicing and

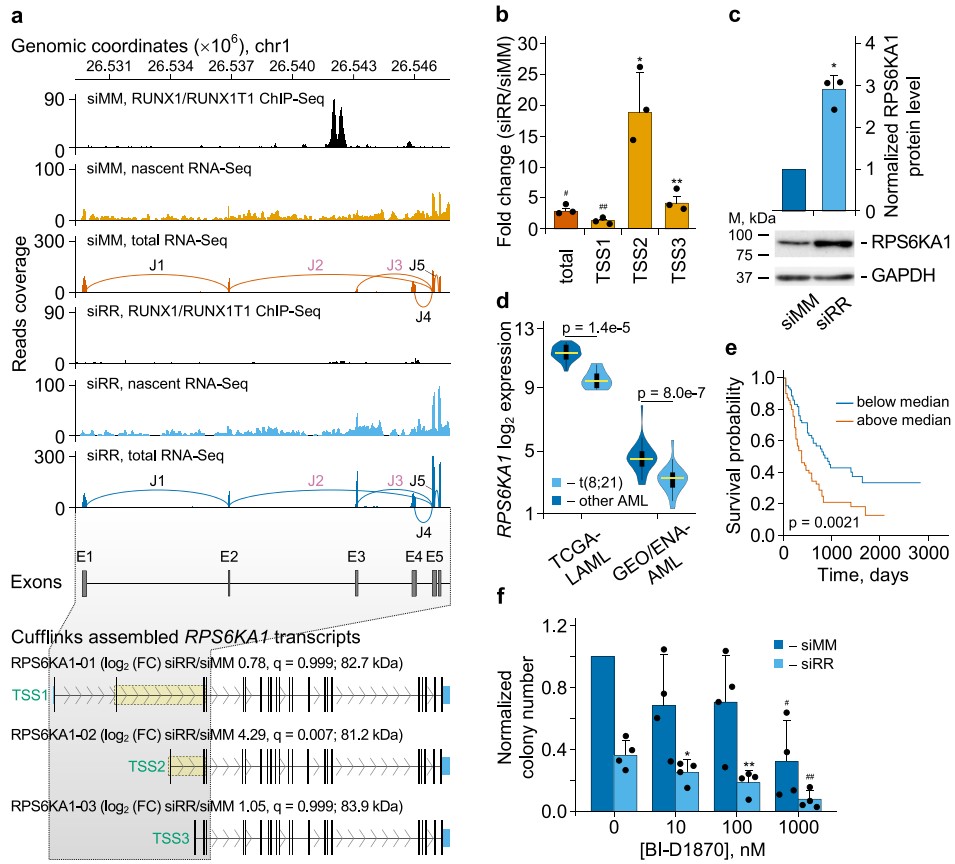

**Fig. 8 Knockdown of *RUNX1/RUNX1T1* leads to differential splicing of *RPS6KA1* transcripts in Kasumi-1 cells. a** IGV screen shot showing the RUNX1/ RUNX1T1 binding and RNA reads from nascent and total RNA-Seq at the 5′-terminal region of *RPS6KA1* locus for siMM-treated and siRR-treated Kasumi-1 cells. E, exons; J, exon-exon junctions. Bottom scheme depicting *RPS6KA1* transcripts. diffEEJs are boxed and highlighted in yellow. **b** qPCR-based quantitation of the change in the activity of various transcription start sites of *RPS6KA1* gene following *RUNX1/RUNX1T1* knockdown. The overall expression level ("total") of *RPS6KA1* gene is also indicated. *$p = 0.029$, **$p = 0.025$, #$p = 0.022$, and ##$p = 0.014$ with one-tailed one sample Student's t-test. Bars represent mean ± SD of three independent experiments. **c** Knockdown of *RUNX1/RUNX1T1* leads to statistically significant ($p = 0.00504$ (*)) changes in the RPS6KA1 protein level in leukemic cells. *P*-value was calculated using one-tailed one sample Student's t-test with $\mu_0 = 1$. Bars represent mean ± SD of three independent experiments. **d** Differential expression of *RPS6KA1* gene in t(8;21) positive and negative AMLs. Gene expression levels were taken from the TGCA-LAML ($n = 173$) and GEO/ENA-AML ($n = 80$) datasets. For the TGCA-LAML dataset, RSEM values of normalized gene expression were used. Normalized expression of genes from GEO/ENA-AML dataset was calculated as RPKM. In each violin plot, horizontal yellow line represents the median of expression distribution, box shows the interquartile range, and whiskers are the minimum and maximum. *P*-values were calculated with two-sided Mann–Whitney U test. **e** Survival of AML patients depending on expression of *RPS6KA1* gene. This survival plot is based on TCGA-LAML dataset ($n = 173$). The dependence of patient survival on gene expression was calculated according to the Cox proportional hazards regression model. **f** Impact of RPS6KA1 inhibition on clonogenicity of t(8;21) AML cells. Kasumi-1 cells were electroporated with the indicated siRNAs followed by plating into semisolid medium containing the indicated concentrations of RPS6KA1 inhibitor. Data was normalized relative to clonogenicity of the siMM electroporated and BI-D1870 untreated leukemia cells. Bars represent mean ± SD of four independent experiments. *$p = 0.053$, **$p = 0.012$, #$p = 0.008$, and ##$p = 0.001$ with one-tailed Student's t-test against respective baseline in untreated cells.

mRNA surveillance. Consequently, factors can be classified according to their response to RUNX1/RUNX1T1 status. One group comprises RBFOX2 and several hnRNPs (HNRNPDL, HNRNPLL, and HNRNPR) with changed expression levels upon RUNX1/RUNX1T1 knockdown, while other hnRNPs such as HNRNPA3, HNRNPF, and HNRNPH1 were not affected (Supplementary Data 11).

Based on the above-mentioned predictive model (see section "Modulation of local epigenetic marks by RUNX1/RUNX1T1 influences alternative splicing"), we developed a short list of sequence features that represent the most important predictors of whether an exon-exon junction is differential or not. This short list included motif frequency and strength for HNRNPA3, MBNL1, PTBP1, RBFOX2, SRSF7, and YBX1 splicing factors (Supplementary Data 12). Partial dependence profiling indicated a strong non-linear relationship between class probability of EEJs

and frequency or strength of splicing regulator motifs. Motifs for all splicing factors of this list similarly affected the discrimination between differential versus non-differential splicing events. For example, there is a positive link between the motif strength of the YBX1 splicing factor in the 3′-end of the intron and the probability of diffEEJs, and a mainly negative relationship between SRSF7 motif score in the downstream exon and differential splicing (Fig. 9e).

To further validate above mentioned findings, we also conducted a motif analysis based on several independent splicing factors binding datasets[53–55]. This analysis confirmed an association between motifs for some splicing factors and differential splicing in the transcriptome of Kasumi-1 cells following RUNX1/RUNX1T1 knockdown (Supplementary Data 13). Moreover, our experimental data also indicate the existence of such a relationship. For instance, the two t(8;21)

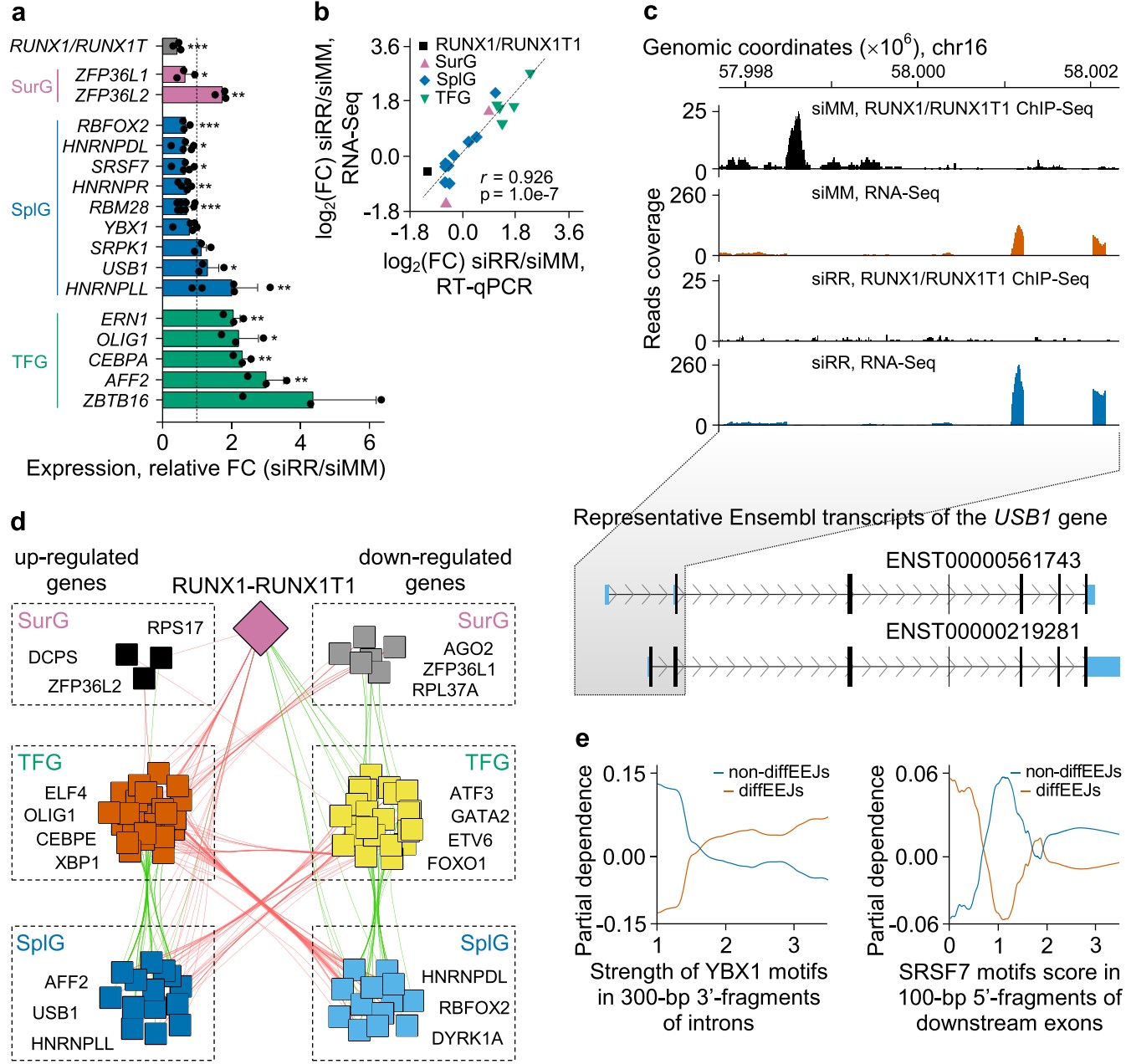

**Fig. 9 RUNX1/RUNX1T1 indirectly affects differential splicing in Kasumi-1 cells. a** Bar graph showing the impact of RUNX1/RUNX1T1 knockdown on the expression of mRNA surveillance genes (SurG), genes encoding splicing factors (SplG) and transcription factors genes (TFG) associated with splicing. Bars represent mean ± SD of three independent qPCR experiments. *$p < 0.05$, **$p < 0.01$, ***$p < 0.001$ with one-tailed Student's t test. **b** Graph showing correlation between RNA-seq and q-PCR validation of differential RNA processing gene expression. **c** IGV screen shot depicting RUNX1/RUNX1T1 binding and RNA-Seq reads for the *USB1* locus. **d** Cytoscape scheme showing RUNX1/RUNX1T1-centered gene regulatory network of genes encoding splicing regulators and mRNA surveillance factors. Terms "upregulated genes" and "downregulated genes" refer to a change in the expression of the genes after *RUNX1/RUNX1T1* knockdown. Positively and negatively co-expressed genes linked by green and red edges, respectively. Examples of the genes from each group are shown in the respective dashed boxes. **e** Effect of the strength of the motifs to splicing factors on the probability of non-diffEEJs and diffEEJs in leukemia cells. These partial dependence plots are based on the results from 1,000 independent runs of the random forest meta-classifier and 1000 classification trees per random forest per run.

AML cell lines Kasumi-1 and SKNO-1 express a non-canonically splice variant of TRAPPC2L lacking exon 4 (Supplementary Fig. 6)[56,57]. TRAPPC2L is not a direct target of the fusion protein, but eCLIP data from K562 cells suggest that SRSF7 binds to all exons of TRAPPC2L (Supplementary Fig. 6g). Interestingly, while knockdown of RUNX1/RUNX1T1 abolishes skipping of exon 4, depletion of SRSF7 has the opposite effect. In

contrast, RBFOX2 knockdown did not affect exon 4 skipping (Supplementary Fig. 6h, i). These combined findings suggest that RUNX1/RUNXT1 corrupts splicing by interfering with SRSF7 function.

Finally, to assess the involvement of RNA surveillance factors in differential splicing, we compared the distribution of RNA isoform expression with and without premature translation

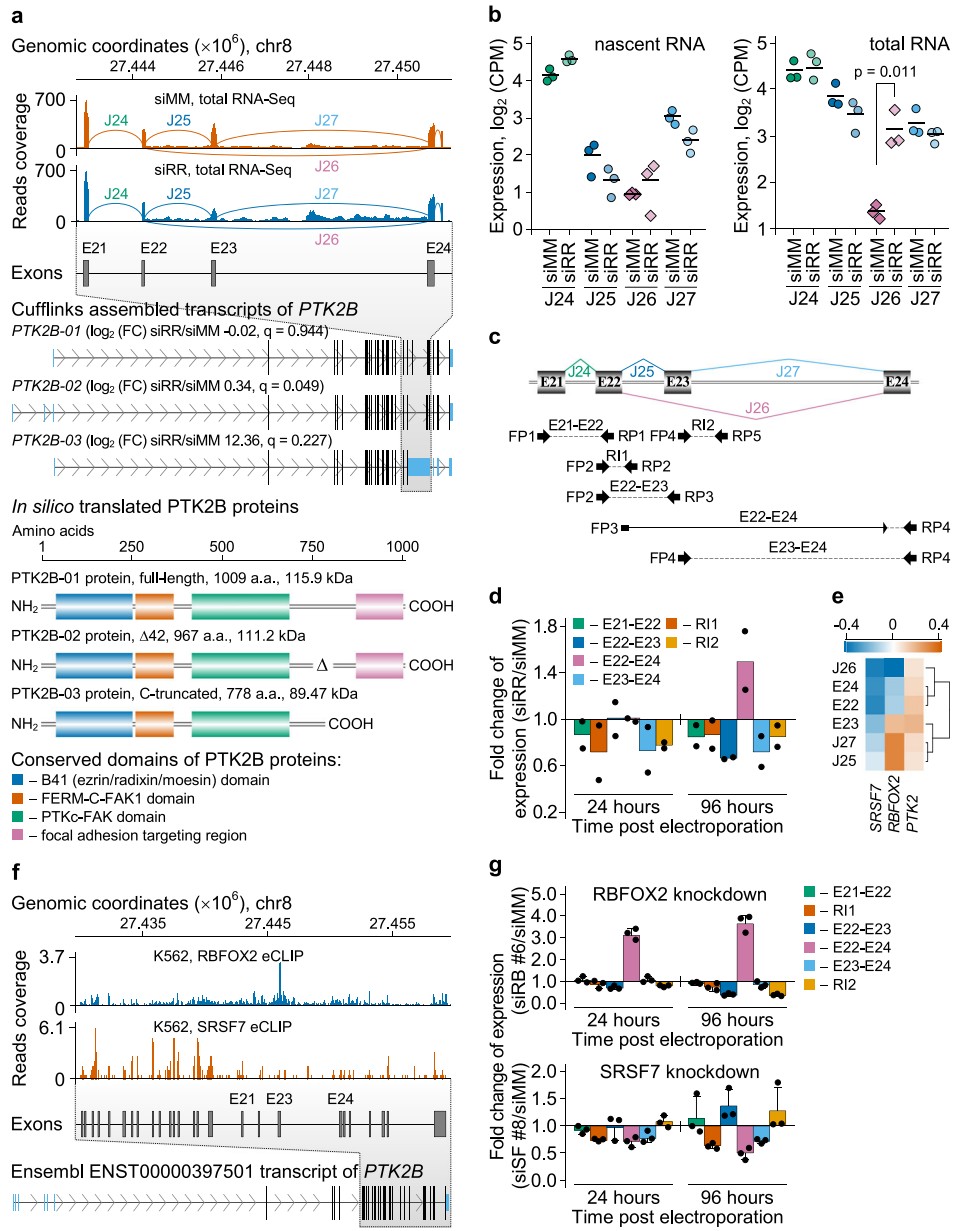

**Fig. 10 RUNX1/RUNX1T1 controls alternative splicing of *PTK2B* transcripts in t(8;21)-positive leukemia cells. a** Splicing graphs, representative Cufflinks assembled transcripts and in silico translated proteins of *PTK2B* gene. This panel is based on Kasumi-1 total RNA-Seq data. Exons and exon-exon junctions are designated by the letters E and J, respectively, and numbered. Differentially used junction J26 is highlighted in purple. For each transcript, Cuffdiff based log₂ fold changes in expression and respective *q*-values are indicated in parentheses. a denotes amino acids. **b** Strip charts demonstrating normalized expression of exon-exon junctions J24 to J27 in Kasumi-1 cells. Genomic location of junctions is shown in **a**. Horizontal lines are arithmetic means. *P*-value was calculated with two-sided Student's t test. **c** and **d** qPCR-based validation of the differential splicing of *PTK2B* transcripts under two siRNA treatment conditions. Binding positions of primers and structure of the expected amplicons are shown in **c**. Bars in **d** represent mean of two independent qPCR experiments. **e** Pearson correlation heatmap demonstrating relationship between expression of the *SRSF7*, *RBFOX2*, or *PTK2* genes and splicing events in the *PTK2B* gene. This map is based on TCGA-LAML dataset (*n* = 173). **f** RBFOX2 and SRSF7 proteins bind *PTK2B* gene. This genome browser snapshot is based on ENCODE eCLIP data. **g** Effect of the siRNA-mediated knockdown of *RBFOX2* and *SRSF7* expression on splicing of the *PTK2B* gene in Kasumi-1 cells. Bars represent mean ± SD of three independent experiments.

termination codons for all the genes with differential splicing. This analysis did not reveal significant differences between the two pools of RNA molecules: about 11% of differentially spliced RNA isoforms contains premature stop codons compared to 14% of non-differential RNA isoforms (odds ratio 0.81, *p*-value = 0.470 according to Fisher's exact test).

Taken together, these findings suggest that besides regulating differential splicing via direct binding and altering chromatin structure, RUNX1/RUNX1T1 dysregulates this process by altering the expression of splicing factors.

**RUNX1/RUNX1T1 regulates alternative splicing of *PTK2B* via RBFOX2.** The *PTK2B* gene (also known as *PYK2*) encodes the focal adhesion kinase 2B, which regulates cell adhesion and migration, two key processes in leukaemogenesis[58]. High

expression levels of this gene are indicative of an inferior clinical outcome (Supplementary Fig. 7a). The gene is subject to alternative splicing: differential splicing of exon 23 results in two protein isoforms of 1009 and 967 amino acids, with the shorter one lacking a nuclear/cytoplasmic localization region between the kinase and the C-terminal focal adhesion targeting domain (Fig. 10a)[59]. Analysis of nascent and total RNA-Seq data from Kasumi-1 cells indicated that *RUNX1/RUNX1T1* knockdown led to increased exon 23 skipping and concomitant joining of exon 22 to exon 24 (Fig. 10b–d). Although RUNX1/RUNX1T1 occupies the *PTK2B* locus at several sites, formation of this alternative junction J26 is unlikely to be directly regulated by the fusion protein, because its closest binding site locates more than 35 kb upstream of 5' splice site of junction J26. Since loss of RUNX1/RUNX1T1 binding was also associated with a more than twofold reduced expression of *RBFOX2* (Fig. 9a), we examined a potential of this regulator of alternative splicing in the retention of exon 23. Analysis of TGCA-LAML and GEO/ENA-AML datasets established a negative correlation between *RBFOX2* expression and exon 23 skipping (Fig. 10e; Supplementary Fig. 7b, c). Notably, a UGCAUG consensus site for RBFOX2 binding is located just 12 nt downstream of this exon and evaluation of ENCODE eCLIP data for the CML cell line K562 revealed direct association of RBFOX2 to exon 23 of *PTK2B* (Fig. 10f).

To examine the impact of RBFOX2 on exon 23 skipping more directly, we perturbed RBFOX2 expression using two independent siRNAs. Either siRNA achieved with 2.5-fold to 4-fold reduced *RBFOX2* levels a similar reduction as RUNX1/RUNX1T1 knockdown (Supplementary Fig. 7d). This loss of *RBFOX2* led to a threefold decreased exon 23 retention again similar to that achieved by depletion of the fusion protein (Fig. 10g, upper panel). Notably, knockdown of SRSF7 did not affect exon 23 retention (Fig. 10g, bottom panel). These combined data show that RUNX1/RUNX1T1 affects alternative splicing of PTK2 by modulating the expression of RBFOX2.

## Discussion

Analyzing modes of malignant dysregulation of the transcriptome usually separates transcriptional from posttranscriptional control mechanisms. Consequently, alterations in splicing patterns have been mainly explored in conjunction with mutations in splicing factors, while investigations regarding mutated or aberrantly expressed transcription factors focused on their impact on transcription[6,17]. However, most of the latter studies include genome wide transcriptomic analyses such as total RNA-Seq as well[21], which are affected by post-transcriptional RNA processing including altered splicing or degradation[49]. Here, we have extended such transcription-centered studies by analyzing the impact of RUNX1/RUNX1T1, a prototype of a leukemic transcriptional regulator, on the generation of alternative transcripts and found that it controls transcription start site and exon choice. Such changes can be associated with altered protein sequences including the generation of isoforms of yet unknown function as shown here for PARL. We find that RUNX1/RUNX1T1 controls RNA-isoform expression by preventing alternative initiation of transcription and, thus, directly link an initiating and driving oncogenic event with the recently discovered pervasive regulation of transcription through alternative promoters in cancer cells[11]. Our work provides an explanation for potential discordances between changes in transcript and protein levels following *RUNX1/RUNX1T1* knockdown.

The precise identification of differential splicing events using RNA-Seq data is not a trivial undertaking; there is no universal and perfectly reliable bioinformatics tool for this task[29,30,60]. For this reason, we employed several algorithms to detect a comprehensive set of differential splicing events at exon or exon–exon junction levels with subsequent experimental validations. Using this approach, we identified 378 genes with significantly different RNA splicing maps in Kasumi-1 cells following the fusion oncogene knockdown. A part of the identified differential splicing events arises from the differential usage of alternative transcription starts and is mediated by fusion protein binding. This result agrees with the observation that transcription factors regulate alternative splicing by occupying and modulating alternative promoters[61–63].

It was recently suggested that RUNX1/RUNX1T1 may recruit the splicing of regulatory proteins including proteins with poly(A) RNA binding activity (i.e., CIRBP, DKC1, GAR1, or RBM3)[15]. These data imply a role of the fusion protein in the control of polyadenylation. Indeed, we found 614 3'-terminal exons that overlap with RUNX1/RUNX1T1 binding peaks (data not shown). However, only 4 of these exons were differentially used in the transcriptome of the siRR-treated versus siMM-treated Kasumi-1 cells (at least twofold changes, $q < 0.0001$). Thus, RUNX1/RUNX1T1 does not substantially affect the selection of mRNA 3'-termini.

In addition, we found a sub-set of splicing-related genes that were differentially expressed upon RUNX1/RUNX1T1 knockdown. This sub-set includes genes encoding classical splicing regulators (for example, *RBM28, RBFOX2, HNRNPLL*), as well as multifunctional genes (for instance, *DYRK1A*) encoding proteins with different activities, including components of the nuclear speckles, which are considered an area of intensive splicing[25,26]. According to our data mining and motif analysis results, these regulatory proteins may additionally contribute to differential splicing in Kasumi-1 cells following the *RUNX1/RUNX1T1* knockdown.

In conclusion, we have shown here that RUNX1/RUNX1T1 regulates alternative RNA splicing for a sub-set of genes in t(8;21)-positive AML, by controlling the choice of transcriptional start sites and by modulating expression of splicing components. These findings add another layer of dysregulation of gene function by nuclear oncoproteins promoting leukaemogenesis.

## Methods

**Cell lines.** The t(8;21)-positive AML cell lines Kasumi-1 (DSMZ no. ACC 220) and SKNO-1 (DSMZ no. ACC 690) were obtained from the DSMZ (LGC Standards GmbH, Wesel, Germany) and cultivated in RPMI1640 containing either 10% fetal bovine serum (Kasumi-1) or 20% fetal bovine serum and 7 ng/ml GM-CSF (SKNO-1).

**siRNA transfections.** Kasumi-1 and SKNO-1 cells ($5 \times 10^5$ cells/ml) were transfected with 200 nM siRNA in standard culture medium at 330 V (Kasumi-1) or 350 V (SKNO-1) for 10 ms using 4 mm electroporation cuvettes and a Fischer EPI 2500 electroporator (Fischer, Heidelberg, Germany)[64]. After 15 min at room temperature, cells were diluted in standard medium to a concentration of $5 \times 10^5$ cells/ml. In all the RUNX1/RUNX1T1 knockdown experiments, the previously established anti-RUNX1/RUNX1T1 active siRR and mismatch control siMM were used[28]. All siRNAs used in this study are listed in Supplementary Table 2.

**Extraction of total cellular RNA, synthesis of the first strand of cDNA and qRT-PCR.** Total cellular RNA was extracted using the RNeasy Mini Kit (QIAGEN GmbH, Hilden, Germany) according to the manufacturer's protocol. Synthesis of the first strand of cDNA was performed from 1 µg of total cellular RNA in 20 µl volume using oligo(dT)$_{18}$ primer and SuperScript$^{TM}$ III Reverse Transcriptase Kit (Thermo Fisher Scientific, Carlsbad, USA) according to the manufacturer's protocol. Quantitative real-time PCR was carried out in triplicates on StepOnePlus Real-time PCR System (Thermo Fisher Scientific, Carlsbad, USA) using Quanti-Tect® SYBR® Green PCR Kit (QIAGEN GmbH, Hilden, Germany) according to the manufacturer's protocol. All the primers (Supplementary Data 8 and Supplementary Data 14) for real-time qPCR were designed using Primer-BLAST on-line tool[65].

**Extraction of total cellular proteins and Western blotting.** Total cellular proteins were extracted simultaneously with the RNeasy Mini Kit-based purification of

RNA by precipitating the RNeasy flowthrough with 2 volumes of acetone[21]. Pelleted proteins were dissolved in urea buffer (9 M urea, 4% CHAPS (3-((3-cholamidopropyl) dimethylammonio)-1-propanesulfonate), 1% DTT) and protein concentrations were measured by Bradford assay.

Western blotting was carried out according to the previously described protocol[29]. Rabbit polyclonal anti-RUNX1 (1:1000; cat. #4334S, Cell Signaling Technology), rabbit monoclonal anti-RPS6KA1 (1:1000; cat. #8408S, Cell Signaling Technology), mouse monoclonal anti-LIMS1 (1:1000; cat. #11890S, Cell Signaling Technology), and mouse monoclonal anti-GAPDH (1:10,000; cat. #AM4300, Invitrogen™) antibodies were used as primary antibodies. Moreover, two different rabbit polyclonal antibodies with C-terminal epitopes were used to detect the PARL proteins: the first one (α-PARL #1; 1:500) was obtained from Thomas Langer' laboratory[44] and the second one (α-PARL #2) was from Abcam (1:1000; cat. #ab45231). In addition, goat anti-mouse (1:10,000; cat. #P0447, Agilent) or anti-rabbit (1:10,000; cat. #sc-2004, Santa Cruz Biotechnology) polyclonal immunoglobulins conjugated with horseradish peroxidase were used as secondary antibodies.

**Nascent RNA-seq**. Procedure for nascent RNA isolation, preparation of RNA-Seq libraries, and massively parallel sequencing was previously described[66]. Briefly, $10^8$ Kasumi-1 cells were treated with 500 µM 4-thiouridine for 1 h. Cells were lysed using TRI Reagent® (Sigma-Aldrich, St Louis, USA) and RNA was purified according to the manufacturer's instruction. Next, 4-thiouridine-incorporated RNA was biotinylated by labeling with 1 mg/ml Biotin-HPDP (Abcam, Cambridge, UK) for 90 min at room temperature. Following chloroform extraction, labeled RNA was separated using magnetic streptavidin beads, beads were washed, and RNA was eluted in two rounds of elution with 100 µl 100 mM DTT. Finally, RNA was purified using RNeasy MinElute Cleanup Kit (QIAGEN GmbH, Hilden, Germany) and samples were sequenced using Illumina HiSeq™ 2000 paired-end sequencing.

**Publicly available high-throughput datasets used in the study**. Our public microarray dataset included data on 106 primary samples of bone marrow or peripheral blood from patients with t(8;21)-positive AML, and 85 samples of bone marrow-derived CD34-positive cells (BM), 32 samples of bone marrow mononuclear cells (BM-MNC) and 100 samples of peripheral blood mononuclear cells (PB-MNC) from healthy donors (Supplementary Data 1). CEL files of the selected microarrays were downloaded from the NCBI GEO repository[67], converted into CHP files, and primary data matrix was corrected against the background, log2-transformed and quantile normalized by the robust multi-array averaging algorithm using Affymetrix Expression Console™ (Thermo Fisher Scientific, Carlsbad, USA).

The subsequent datasets were downloaded from European Nucleotide Archive as FASTQ files generated on the Illumina HiSeq™ 2000/2500 platform. Datasets comprised RNA-Seq data on primary samples from 20 patients each with either t(8;21)-positive AML, karyotypically normal AML (KN), inv(16)-positive AML and different rearrangements of the MLL gene (MLLr). Furthermore, they comprised 20 samples each of normal bone marrow-derived CD34-positive cells (BM), umbilical cord blood-derived CD34-positive cells (CB) and mobilized normal peripheral blood-derived CD34-positive cells (PB) (Supplementary Data 1).

**Annotated public datasets**. All annotations of human genome were downloaded from Ensembl database in GTF/GFF format[32]. We used Ensembl release 85 based on GRCh38.p7 reference assembly of human genome. Genomic coordinates of the CpGs islands in the human genome were downloaded via FTP server of UCSC Genome Browser[68]. The whole set of GenBank human mRNA and ESTs sequences was downloaded via FTP server of UCSC Genome Browser in GTF format. We used sequences that were pre-aligned against the GRCh38/hg38 reference assembly of the human genome with BLAT[69]. Any sequences with mismatches and less than four aligned blocks were removed. In addition, sequences with the exon length <25 bp and intron length <50 bp were filtered out.

**Statistical analysis**. Statistical analyses for each experiment were performed as described in the text body or in the corresponding figure legends. Data between groups were compared, unless otherwise specified, using a two-sided Student's t-test, Mann–Whitney U test, $\chi^2$-test or Fisher' exact test. All data are presented as mean ± SD. Differences between groups were considered statistically significant at $p < 0.05$.

**Reporting summary**. Further information on research design is available in the Nature Research Reporting Summary linked to this article.

## Data availability

Nascent RNA next-generation sequencing has been submitted to the Gene Expression Omnibus (accession number GSE160792). Total RNA-Seq, ChIP-Seq, and DNase-Seq samples were previously deposited in the Gene Expression Omnibus (accession numbers GSE54478 and GSE29222). Publicly available high-throughput datasets used in the study are described in Supplementary Data 1. All other relevant data supporting the key

findings of this study are available within the article, supplementary information or from the corresponding authors upon request. Source data are provided with this paper.

## Code availability

All R codes of this study were deposited at GitHub software project TranscriptomicFeatures (https://github.com/VGrinev/TranscriptomicFeatures)[70]. These codes are freely available under GNU General Public License v3.0.

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

## Acknowledgements

The authors thank Dr. Aliaksandra Radzisheuskaya (Cell Biology Program and Center for Epigenetics, Memorial Sloan Kettering Cancer Center, New York, USA) for carefully reading and improving the manuscript, Dr. Petr Nazarov (Luxembourg Institute of Health, Strassen, Luxembourg) for excellent practical ideas with independent component analysis, Dr. Paul Boutz (Koch Institute for Integrative Cancer Research, Massachusetts Institute of Technology, Cambridge, USA) for comprehensive discussion of the pipeline and tips on identifying retained introns, and Dr. Thomas Langer (University of Cologne, Cologne, Germany) for providing us with original anti-PARL antibodies. Research in the V.V.G. laboratory was supported in part by the Ministry of Education of the Republic of Belarus, grant #3.08.3 (947/54, 469/54). O.H. is supported by grants from Cancer Research UK (C27943/A12788), Bloodwise (15005), Kay Kendall Leukemia Fund (KKL1142), the North of England Children's Cancer Fund and KIKA (329). C.B. is supported by Bloodwise (15001).

## Author contributions

V.V.G., F.B., I.M.I., R.C., J.S., A.v.O., M.S., H.M., M.R., N.M.-S., and S.A.A. performed experiments. V.V.G., I.M.I., S.N., and S.A.A. analyzed data. T.V.R. and F.B. contributed to preparation and editing of the manuscript. V.V.G. and O.H. conceived the study and, together with C.B., supervised experiments and wrote the manuscript.

## Competing interests

The authors declare no competing interests.
