## [Peer Review File · Nature Communications]

Reviewers' comments:

Reviewer #1 (Remarks to the Author):

In their manuscript, Heidenreich and colleagues, address a vexing issue in leukemia research. They investigated alternative splicing by one of the most frequently detected fusion oncogenes in acute myeloid leukemia - RUNX1-RUNX1T1. By doing this, they found that RUNX1-RUNX1T1 induces alternative splicing via two mechanisms: alternative transcription start sites selection and direct or indirect control of the expression of splicing factors. Thereby RUNX1-RUNX1T1 affects several functional groups of genes and produces unique proteins, proposing alternative splicing as one oncogenic mechanism of RUNX1-RUNX1T1. As alternative splicing was shown in other contexts to be a crucial oncogenic event and as several splicing factors were found to be mutated in different types of cancer, including myeloid malignancies, the article touches a current hot topic in cancer research.

The article is therefore interesting. The authors applied numerous and state of the art bioinformatic methods to (previously published) data sets from one cell line (Kasumi-1) with/without knockdown of the fusion oncogenes (RNA-Seq, nascent RNA-seq, ChIP-Seq) and patient samples to rigorously make their conclusion and make novel findings.

The article can be further improved by addressing major points.

1. The article is extremely complicated and difficult to read. Abbreviations are inflationary used and sometimes not even introduced. Abbreviations should be reduced to a minimum. Different methods for isoform prediction or quantification are presented and compared against each other, without even introducing the method per se, the advantage of each method and differences between them (e.g. Fig. 1c or 1g-h and multiple other figure). The figures are unclear without reading the figure legends (e.g. Fig. 1 e-I, Fig. 2a-h etc.). It is unclear which samples were used and also which genes were included in the analyses. The subsection "differential expression of splicing factors..." (page 10) is completely disconnected from the rest of the paper. E.g. the authors write "Interestingly, we and others have previously shown that t(8;21)-positive leukemia cells have a specific signature of genes encoding splicing factors and mRNA surveillance genes expression.". This is exactly what is shown in Fig. 1a and the first subsection. Hence, the clarity and red line of the manuscript needs substantial improvements.

2. Many interesting hypotheses were generated by bioinformatic predictions and analyses, mainly of published datasets. However, key findings should be experimentally validated. E.g. differential PARL isoform usage seems to be specifically linked to t(8;21) leukemia in comparison to normal CD34-positive cells. Does knockdown of PARL-02 change the phenotype of Kasumi-1 cells or does overexpression of PARL-02 induce proliferation or differentiation arrest in CD34-positive cells? Does the inverse apply to the two other isoforms (PARL-06 and PARL-08). Showing that would greatly increase the impact of the findings. The same applies to RPS6KA1 and LIMS1 isoforms.

3. Similarly, the authors propose that RUNX1-RUNX1T1 can indirectly affect splicing and mRNA processing by regulating splicing factors. The gene set enrichment analysis presented in Fig. 1a looks odd. Although the values imply highly significant downregulation of genes associated with mRNA processing and splicing, the enrichment plot suggests that the genes cluster in the middle of the rank list; i.e. not significantly changed. The authors should explain this discrepancy and verify the downregulation of crucial splicing factors via RT-qPCR. In addition, the author should verify alternative splicing of candidate genes upon knockdown of the top differentially expressed splicing factors from their analysis in Kasumi-1 cells or CD34-positive cells.

Reviewer #2 (Remarks to the Author):

Grinev et al. describe defects in alternative splicing in t(8;21)-positive AML. By comparing cells with and without RUNX1/RUNX1T1 knockdown, they identify differentially expressed genes and differentially spliced exons. The authors integrate a large number of published datasets to identify changes in alternatively spliced isoforms in cancer patients. They identify changes in

RUNX1/RUNX1T1 binding in alternatively spliced genes, especially in those with alternative first exon usage. In addition, they uncover changes in splicing-factor expression that could potentially explain differential splicing. Finally, they identify map splicing events that are likely to impact known protein domains.

Findings from this study will advance our understanding of alternative splicing misregulation in tumors, by linking an aberrant transcription factor with splicing defects. However, the current draft is not ready for publication. A number of critical questions have to be addressed before this study can be published.

MAJOR POINTS

1) Critical experimental details are missing throughout the manuscript thus making it difficult to understand what is actually being analyzed, why, and to interpret the conclusions of the analysis. This needs to be addressed in the main text. For example:

- Page 4: "We initially analyzed published data sets (Supplementary Table 1) for a potential association between RUNX1/RUNX1T1 and RNA processing"- what type of datasets and analysis is being performed? How are significant genes called?
- Page 5: "The differential splicing events were detected at the level of both exon usage and exon-exon junctions (EEJs) using DEXSeq, limma/diffSplice and JunctionSeq". These methods are quite different, how are the significant events filtered and compared?
- Page 6: "diffEEJs identified by both diffSplice and JunctionSeq were confirmed using qPCR (Supplementary Table 6)." How many events are tested vs. validated? How are these selected? Which one are considered validated in Table 6.
- Page 6: "specific retained introns a trend towards changed expression after RUNX1/RUNX1T1 knockdown (Supplementary Fig. 1d, e, f, g, h)." – what is that trend? Are they more retained? How many introns are affected? Few introns seem to have a logFC on Fig S1.
- Page 11: "By combining of different approaches [...]"- which approaches?
- Supplementary Table 3 reports retained introns detected in the transcriptome of Kasumi-1 cells in two categories "both" and "knockdown" – what exactly is "both"? both siRNA compared to what exactly? Splicing events detected in cells with siMM should be filtered out as non differential, please explain what the control is otherwise?

2) Data supporting key statement and findings is not shown. This lack of transparency is not acceptable, the readers should be able to see that data to judge by themselves. For example:

- Page 5: "Furthermore, RUNX1/RUNX1T1 binding sites are often present within the genes encoding splicing associated factors or in their immediate vicinity." – where is this data?
- Page 6-7 "we detected about 80% of Kasumi-1-associated diffEEJs that are abundant in the transcriptome of primary t(8;21)-positive leukemia blasts (Fig. 2f)." and "three independent components (subsets), or simply ICs, of EEJs that clearly separate t(8;21)-positive AML cells from normal CD34-positive cells (Fig. 2g)" – where is a list of these EEJs? How many EEJs are detected in patients? Which EEJs are these?
- Page 9: "In our dataset, we found that 38% of genes with differential splicing are also enriched for the fusion protein binding sites (data not shown)." – where is this data?
- Page 12: "Motifs for all the other splicing factors from the above list similarly affected the discrimination between differential versus non-differential splicing events (data not shown)." – where is this data?
- Page 13: "This analysis revealed no significant difference between the wild type and knockdown conditions (data not shown)." – where is this data?

3) A major concern is the multitudes of computational methods used to analyze data in the paper, and the lack of explanation why one method is used vs. the other. We agree that there is currently no standard pipeline for splicing analysis, and that each method is better at detecting certain types of events. The authors therefore use different methods to identify splicing. Not surprisingly, the overlap between the methods is limited, as described before. The author claim "This is due to the differences in the pre-processing, normalization and transformation of the data, as well as in the use of different statistical models at the step of identification of diffEEJs by the algorithms."; yet what is missing is the description of the significance cut-offs applied for each method? How are significant changes defined? How many reads are required to support inclusion vs. skipping? The

splicing analysis is overall extremely convoluted, with no clear explanation on what is compared. On page 13 the authors “used the Cufflinks-assembled transcriptome of Kasumi-1 cells to select all 762 transcripts of the genes affected by RUNX1/RUNX1T1-dependent differential splicing. These genes gave rise to 214 differentially spliced and 548 non-differentially spliced RNA isoforms.” – it is unclear if the authors are referring to 762 or 214 differentially spliced isoforms? What are the 548 non-differentially spliced RNA isoforms?

Furthermore, The authors seem to be cherry picking which method they use to generate gene list; for example page 13 “we used the Cufflinks-assembled transcriptome of Kasumi-1 cells to select all 762 transcripts of the genes affected by RUNX1/RUNX1T1-dependent differential splicing.” while on page 6 “we identified 395 differentially used exons [...] over 275 individual genes” and “ 177 differential EEJs (diffEEJs) [...] over 150 genes”, whereas figure 1 c shows 326 isoforms overlapping between the two detection methods. Again, we understand the need to use different methods, but these need to be explained and properly compared. Are we combining the results of both methods, or only looking at overlap. How many of these events have been validated by RT-PCR?

Finally, a major criticism of the used methods is the lack of de novo transcriptome reconstruction, and the reliance on reference transcriptome. Indeed, cancer cells have been shown to express novel isoforms, and these could be absent in the reference transcriptome. This should at least be discussed.

4) All the data is supported by one siRNA in one cell line. The authors should acknowledge this limitation. Also please show whether RUNX1/RUNX1T1 affects cell viability. Differentially splicing changes have been previously associated with cell death, therefore it is critical to show that the changes detected here are not a mere reflection of cell dying.

5) A recent study (Yoshimi et al. Nature 2019.) uncovered combination of mutations in splicing factors, e.g., SRSF2, and other genomic alterations in AML using RNA-seq datasets. Do any of these datasets analyzed here have a mutation in a splicing factor which would explain splicing changes?

6) The authors should be careful when inferring causality. For example on page 5 “These findings strongly support a regulatory role of this leukaemic fusion protein in mRNA splicing and predict changes in splicing pattern in response to perturbing RUNX1/RUNX1T1 activity.” This effect could be due to direct changes in RUNX1/RUNX1T1 binding, as well as indirect for example by changing splicing factor expression. To prove direct binding the authors should prevent/abolish binding at a specific site and demonstrate a change in splicing. Also on page 17: “In conclusion, we have shown here that RUNX1/RUNX1T1 affects alternative splicing of RNA in AML [...] and by modulating expression of splicing components.”- Yet the authors do not detect changes in splicing factors that are likely to regulated the spliced events as stated “However, the expression of the SRSF7 gene itself is not altered after the RUNX1/RUNX1T1 knockdown.”. If the author want to link a specific factor to spliced isoforms detected in this study, they could integrate findings from ENCODE cell lines with splicing factor knockdown, for example SRFS7 KD.

7) The authors perform splicing-factor binding analysis using published motifs, yet they fail to include several relevant high-throughput studies, including Ray et al. Nature 2013, Dominguez et al. Mol Cell 2018 and binding data generated by the ENCODE consortium (www.encode.org and <https://doi.org/10.1101/179648>). Moreover, it is unclear from the material and methods whether control exon/intron sequences were randomly selected from the genome, or selected from known alternatively spliced exons that are not differentially spliced in the studied experiment. Since the sequence composition is quite different between constitutive and alternative exons, this is a critical question. Additionally, the material and methods states different methods and threshold were used for the detection and scoring of distinct binding motifs. These differences could introduce detection bias which should be discussed in the text. Whether motifs scored by these different methods can be directly compared as done here, remains to be determined.

MINOR POINTS

1) Please clarify the comment and figure legend on Page 6 " At the same time, no preference was found among the exons of non-coding RNAs (Fig. 1i)."

We found the reviewer's comments very helpful and have addressed them as below:

Reviewer #1 (Remarks to the Author):

In their manuscript, Heidenreich and colleagues, address a vexing issue in leukemia research. They investigated alternative splicing by one of the most frequently detected fusion oncogenes in acute myeloid leukemia - RUNX1-RUNX1T1. By doing this, they found that RUNX1-RUNX1T1 induces alternative splicing via two mechanisms: alternative transcription start sites selection and direct or indirect control of the expression of splicing factors. Thereby RUNX1-RUNX1T1 affects several functional groups of genes and produces unique proteins, proposing alternative splicing as one oncogenic mechanism of RUNX1-RUNX1T1. As alternative splicing was shown in other contexts to be a crucial oncogenic event and as several splicing factors were found to be mutated in different types of cancer, including myeloid malignancies, the article touches a current hot topic in cancer research. The article is therefore interesting.

The authors applied numerous and state of the art bioinformatic methods to (previously published) data sets from one cell line (Kasumi-1) with/without knockdown of the fusion oncogenes (RNA-Seq, nascent RNA-seq, ChIP-Seq) and patient samples to rigorously make their conclusion and make novel findings.

We would like to thank this reviewer for this very positive evaluation!

The article can be further improved by addressing major points. 1. The article is extremely complicated and difficult to read. Abbreviations are inflationary used and sometimes not even introduced. Abbreviations should be reduced to a minimum.

We fully agree with the reviewer and have significantly reduced the amount of abbreviations. In addition, we have included introductions where they were missing in the previous manuscript.

Different methods for isoform prediction or quantification are presented and compared against each other, without even introducing the method per se, the advantage of each method and differences between them (e.g. Fig. 1c or 1g-h and multiple other figure). The figures are unclear without reading the figure legends (e.g. Fig. 1 e-l, Fig. 2a-h etc.). It is unclear which samples were used and also which genes were included in the analyses.

In the first five sections of RESULTS, we focused on the analysis of differential usage of exons or exon-exon junctions following RUNX1/RUNX1T1 knockdown, but not full-length RNA isoforms. A brief introduction to the methods for detecting such splicing events is given in the first section of RESULTS:

“The differential splicing events were detected at the level of both exons and exon-exon junctions (EEJs). Differentially used exons (diffUEs) were identified by use of DEXSeq, limma/diffSplice and JunctionSeq, while differentially used exon-exon junctions (diffEEJs) were identified using limma/diffSplice and JunctionSeq functionality²⁸⁻³⁰. This combined approach detected a comprehensive set of differential splicing events in dependence on the RUNX1/RUNX1T1 status.”

A more detailed description of these methods is now provided in Supplementary Information, where we explain the limma/diffSplice approach in more detail. Since the DEXSeq and JunctionSeq methods are well established algorithms, we introduced them only briefly. However, all respective scripts and in-laboratory developed R codes (with detailed comments) for detection of differential splicing at level of elementary splicing events are deposited on GitHub (<https://github.com/VGrinev/transcriptome-analysis>, sections “Exon-exon junctions” and “Exons”). Furthermore, where necessary, we added figure titles to make them more self-explanatory and amended figure legends to provide essential background information. For instance, an excerpt of the legend of Figure 1 reads now as:

“**c** Distribution of retained introns (RIs) between the transcriptomes of the siRR- and siMM-treated Kasumi-1 cells. The RIs were detected using the DESeq2 and edgeR/limma algorithms. **d** Scatter plot showing the distribution of siRR- and siMM-specific RIs after multidimensional reduction of RIs expression data using t-distributed stochastic neighbor embedding. The RIs were detected using the edgeR/limma algorithm, and very similar results were observed with DESeq2. **e, f** Volcano plots showing differential usage of exons identified by DEXSeq **e** or limma/diffSplice **f** algorithms. Orange and yellow rhombi indicated differentially used canonical exons or RIs, respectively, with more than 2-fold change and $q < 0.1$. **g** Venn diagram showing overlap between differentially used exons (diffUEs) identified by DEXSeq, limma/diffSplice and JunctionSeq algorithms.”

The subsection "differential expression of splicing factors..." (page 10) is completely disconnected from the rest of the paper. E.g. the authors write "Interestingly, we and others have previously shown that t(8;21)-positive leukemia cells have a specific signature of genes encoding splicing factors and mRNA surveillance genes expression.". This is exactly what is shown in Fig. 1a and the first subsection. Hence, the clarity and red line of the manuscript needs substantial improvements.

We have now completely reorganized the manuscript by starting from the observation that the RUNX1/RUNX1T1 gene signature indicates the regulation of RNA splicing and processing by the fusion protein over the analysis of exon usage and intron retainment and its predicted consequences for the composition of proteins to the direct regulation of alternative promoters and the indirect control of alternative splicing. We appreciate that this reviewer might prefer the alternative splicing preceding the alternative transcription starts; however, we finally felt that the alternative start sites as the more direct control would be an easier to follow approach to describe the underlying mechanisms of RUNX1/RUNX1T1-mediated control of mRNA isoform generation.

2. Many interesting hypotheses were generated by bioinformatic predictions and analyses, mainly of published datasets. However, key findings should be experimentally validated. E.g. differential PARL isoform usage seems to be specifically linked to t(8;21) leukemia in comparison to normal CD34-positive cells. Does knockdown of PARL-02 change the phenotype of Kasumi-1 cells or does overexpression of PARL-02 induce proliferation or differentiation arrest in CD34-positive cells? Does the inverse apply to the two other isoforms (PARL-06 and PARL-08). Showing that would greatly increase the impact of the findings. The same applies to RPS6KA1 and LIMS1 isoforms.

We would like to thank the reviewer for this very interesting suggestion. The impact of differential expression of PARL on AML turned out to be complex and will require more in-depth studies to understand it. However, we have performed additional functional studies with PARL and RPS6KA1 (Figures 7, 8 and Supplementary Figure 3) and PTK2B and TRAPPC2L (Figure 10 and Supplementary Figure. 6; see also response to next point) as examples for regulation of alternative TSSs and alternative splicing, respectively. Specifically, we have used a dCas9-KRAB approach to prove the functionality of the alternative TSS in PARL and performed perturbation experiments using an inhibitor of RPS6KA1 in combination with RUNX1/RUNX1T1 knockdown. We also included new data regarding survival and differential expression of isoforms across AMLs. As such we have provided more mechanistic insights into the consequences of alternatively splicing specific transcripts.

3. Similarly, the authors propose that RUNX1-RUNX1T1 can indirectly affect splicing and mRNA processing by regulating splicing factors. The gene set enrichment analysis presented in Fig. 1a looks odd. Although the values imply highly significant downregulation of genes associated with mRNA processing and splicing, the enrichment plot suggests that the genes cluster in the middle of the rank list; i.e. not significantly changed. The authors should explain this discrepancy and verify the downregulation of crucial splicing factors via RT-qPCR. In addition, the author should verify alternative splicing of candidate genes upon knockdown of the top differentially expressed splicing factors from their analysis in Kasumi-1 cells or CD34-positive cells.

We have extended our validation of differentially expressed splicing factors and have also included knockdown experiments targeting SRSF7 and RBFOX2 and their effect on splicing of TRAPPC2L and PTK2 (see Figures 9 and 10, Supplementary Figures 6). We have also replaced some of the GSEAs with new analyses proving a strong enrichment of genes encoding RNA binding proteins and of factors involved in snRNP assembly (see Figure 1).

Reviewer #2 (Remarks to the Author):

Grinev et al. describe defects in alternative splicing in t(8;21)-positive AML. By comparing cells with and without RUNX1/RUNX1T1 knockdown, they identify differentially expressed genes and differentially spliced exons. The authors integrate a large number of published datasets to identify changes in alternatively spliced isoforms in cancer patients. They identify changes in RUNX1/RUNX1T1 binding in alternatively spliced genes, especially in those with alternative first exon usage. In addition, they uncover changes in splicing-factor expression that could potentially explain differential splicing. Finally, they identify map splicing events that are likely to impact known protein domains.

Findings from this study will advance our understanding of alternative splicing misregulation in tumors, by linking an aberrant transcription factor with splicing defects. However, the current draft is not ready for publication. A number of critical questions have to be addressed before this study can be published.

We would like to thank also this reviewer for this very positive general appraisal and we do hope that we have been able to respond satisfactorily to her or his comments.

MAJOR POINTS

1) *Critical experimental details are missing throughout the manuscript thus making it difficult to understand what is actually being analyzed, why, and to interpret the conclusions of the analysis. This needs to be addressed in the main text. For example:*

- Page 4: “We initially analyzed published data sets (Supplementary Table 1) for a potential association between RUNX1/RUNX1T1 and RNA processing” - what type of datasets and analysis is being performed? How are significant genes called?

First, we reanalyzed our previously published ChIP-Seq and total RNA-Seq data sets obtained from siMM- and siRR-treated Kasumi-1 cells. These data sets are listed in Supplementary Table 1 (group “In-laboratory generated data from Kasumi-1 cells”). We applied the DAVID and GSEA algorithms to these data. We have now included this information in the text. We also added a new Supplementary Table 3 with a complete list of direct target genes of RUNX1/RUNX1T1 fusion protein in siMM-treated Kasumi-1 cells. After annotation and rigorous inspection of this table, we selected and entered into the text some interesting examples of significant genes.

- Page 5: “The differential splicing events were detected at the level of both exon usage and exon-exon junctions (EEJs) using DEXSeq, limma/diffSplice and JunctionSeq”. These methods are quite different, how are the significant events filtered and compared?

In the revised version of article, a brief introduction to the methods for detecting differential splicing events is given in the first section of RESULTS. A more detailed description of these methods is provided in Supplementary Methods, where we also describe the limma/diffSplice approach. The DEXSeq and JunctionSeq methods are introduced only briefly because these are well-known published algorithms. All in-laboratory developed R codes (with detailed comments) for detection of differential splicing at level of elementary splicing events are deposited on GitHub (<https://github.com/VGrinev/transcriptome-analysis>, sections “Exon-exon junctions” and “Exons”).

We applied three different filters during the pre-processing of raw count data and identification of the differential splicing events. The first filter was sequencing depth: $FPM \geq 1$ with DEXSeq, $CPM \geq 1$ with limma/diffSplice and the minimum mean normalized read coverage of 10 with JunctionSeq. These sequencing depth (or expression) thresholds were selected on the basis of two criteria: i) localization of the fossa on the distribution curve that separate the noise from a specific signal; ii) robust validation of minimally expressed events by qPCR. The second filter was a fold change in usage of an exon or exon-exon junction between different siRNA treatments. We applied the conservative threshold of $abs|\log_2(FC)_{siRR/siMM}| \geq 1$ to select differentially used splicing events. Finally, the third filter was the statistical significance of the $\log_2(FC)_{siRR/siMM}$ value. We used conservative threshold of 0.1 or 0.01 for false discovery rate adjusted (by classical method of Benjamini Y. and Hochberg Y.) p-value. All these filters are mentioned in the main text, and are also now described in Supplementary Methods.

- Page 6: “diffEEJs identified by both diffSplice and JunctionSeq were confirmed using qPCR (Supplementary Table 6).” How many events are tested vs. validated? How are these selected? Which one are considered validated in Table 6.

In the revised version of article, we selected 17 differentially used exon-exon junctions for qPCR validation. We included in Supplementary Table 7 (former Supplementary Table 6) the results of all validations. The selection of exon-exon junctions for validation was based on two criteria: i) the list of junctions should represent all the main splicing modes (alternative first exon, alternative 5' splice site, alternative 3' splice site, alternative last exon and skipped cassette exon); ii) the list of junctions should include differential events identified by only one of the algorithms or both algorithms.

- Page 6: “specific retained introns a trend towards changed expression after RUNX1/RUNX1T1 knockdown (Supplementary Fig. 1d, e, f, g, h).” – what is that trend? Are they more retained? How many introns are affected? Few introns seem to have a logFC on Fig S1.

In the revised version of article, we clarified the main text about the nature of this trend. In total, we detected 3540 non-overlapping bins of retained introns in control (siMM-treated) and/or experimental (siRR-treated, with RUNX1/RUNX1T1 knockdown) Kasumi-1 cells (Supplementary Table 4). Of all retained introns, 469 and 419 were expressed in either control or knockdown leukemia cells, respectively.

- Page 11: “By combining of different approaches [...]” - which approaches?

We used RNA-Seq and microarray data and three different algorithms (Cufflinks/Cuffdiff, DESeq2 and edgeR/limma) for identification of differentially expressed genes. This information is in Supplementary Table 10 (former Supplementary Table 7). In the revised version of article, the changed text reads as “By combining RNA-Seq (Supplementary Table 1, GSE54478) and microarray (Supplementary Table 1, GSE29223) data and three algorithms (Cufflinks/Cuffdiff, DESeq2 and edgeR/limma), we identified 42 genes encoding splicing factors and 14 mRNA surveillance genes with statistically significant differential expression in Kasumi-1 cells following the RUNX1/RUNX1T1 knockdown (Supplementary Table 10; Fig. 9a, b).”

- Supplementary Table 3 reports retained introns detected in the transcriptome of Kasumi-1 cells in two categories “both” and “knockdown” – what exactly is “both”? both siRNA compared to what exactly? Splicing events detected in cells with siMM should be filtered out as non differential, please explain what the control is otherwise?

The field “Condition” of Supplementary Table 4 (former Supplementary Table 3) reports under which siRNA treatment the retained intron was detected: i) in siMM-treated Kasumi-1 cells only (“control”), ii) in siRR-treated Kasumi-1 cells only (“knockdown”), or iii) under both treatments (“both”). These explanations are provided under Supplementary Table 4 in note 3.

To detect retained introns in the transcriptome of Kasumi-1 cells, we exploited the idea of “null distributions” that was proposed by Phillip Sharp’s group (Boutz et al., 2015; Braun et al., 2017). We analyzed RNA-Seq data separately for siMM- and siRR-treated Kasumi-1 cells. At final step of analysis and for each siRNA treatment, we selected introns that passed a false discovery rate adjusted p-value threshold of 0.01, fold change threshold of 2 (empirical data versus theoretical null distribution) and a required minimum of 20 reads per 100 nucleotides of the effective length of an intron averaged over all the original RNA-Seq samples. These introns were called *in silico* detected retained introns, or simply retained introns. A detailed description of our analysis is given in Supplementary Methods. All our R codes (with detailed comments) for detection of retained introns and differential usage of these genomic elements are deposited on GitHub (<https://github.com/VGrinev/transcriptome-analysis>, section “Retention of introns”).

2) *Data supporting key statement and findings is not shown. This lack of transparency is not acceptable, the readers should be able to see that data to judge by themselves. For example:*

- Page 5: *“Furthermore, RUNX1/RUNX1T1 binding sites are often present within the genes encoding splicing associated factors or in their immediate vicinity.” – where is this data?*

We added a new Supplementary Table 3 with a complete list of direct target genes of RUNX1/RUNX1T1 fusion protein in siMM-treated Kasumi-1 cells.

- Page 6-7 *“we detected about 80% of Kasumi-1-associated diffEEJs that are abundant in the transcriptome of primary t(8;21)-positive leukemia blasts (Fig. 2f).” and “three independent components (subsets), or simply ICs, of EEJs that clearly separate t(8;21)-positive AML cells from normal CD34-positive cells (Fig. 2g)” - where is a list of these EEJs? How many EEJs are detected in patients? Which EEJs are these?*

We prepared two additional supplementary tables (Supplementary Table 8 and Supplementary Table 9) with all necessary statistics.

- Page 9: *“In our dataset, we found that 38% of genes with differential splicing are also enriched for the fusion protein binding sites (data not shown).” – where is this data?*

We included an additional column “Distance to the nearest RUNX1/RUNX1T1 binding peak” to the Supplementary Table 5 and Supplementary Table 6 with the requested data. We also made appropriate changes to the text (see section “Differential splicing can arise from alternative promoter usage”).

- Page 12: *“Motifs for all the other splicing factors from the above list similarly affected the discrimination between differential versus non-differential splicing events (data not shown).” – where is this data?*

We included an additional Supplementary Table 12 with all necessary data.

- Page 13: *“This analysis revealed no significant difference between the wild type and knockdown conditions (data not shown).” – where is this data?*

We included this information to the main text of the article.

“Finally, to assess the involvement of RNA surveillance factors in differential splicing, we compared the distribution of RNA isoform expression with and without premature translation termination codons for all the genes with differential splicing. This analysis did not reveal significant differences between the two pools of RNA molecules: about 11% of differentially spliced RNA isoforms contains premature stop codons compared to 14% of non-differential RNA isoforms (odds ratio 0.81, p-value = 0.470 according to Fisher’s exact test).”

The algorithm for identification of significant open reading frames and premature termination codons in transcripts is described in Supplementary Methods. All relevant R codes (with detailed comments) are deposited on GitHub (<https://github.com/VGrinev/transcriptome-analysis>, section “Functional annotation”).

3) A major concern is the multitudes of computational methods used to analyze data in the paper, and the lack of explanation why one method is used vs. the other. We agree that there is currently no standard pipeline for splicing analysis, and that each method is better at detecting certain types of events. The authors therefore use different methods to identify splicing. Not surprisingly, the overlap between the methods is limited, as described before. The author claim “This is due to the differences in the pre-processing, normalization and transformation of the data, as well as in the use of different statistical models at the step of identification of diffEJs by the algorithms.”; yet what is missing is the description of the significance cut-offs applied for each method? How are significant changes defined? How many reads are required to support inclusion vs. skipping? The splicing analysis is overall extremely convoluted, with no clear explanation on what is compared.

Indeed, we applied three different filters during the pre-processing of raw count data and identification of the differential exon-exon junctions. We used $CPM \geq 1$ for limma/diffSplice and the minimum mean normalized read coverage of 10 with JunctionSeq. These sequencing depth thresholds were selected on the basis of two criteria described above. The second one was a fold change in usage of exon-exon junction between different siRNA treatments. We applied the conservative threshold of $abs|\log_2(FC)_{siRR/siMM}| \geq 1$ to select differentially used splicing events. Finally, the third filter was the statistical significance of the $\log_2(FC)_{siRR/siMM}$ value, and we used conservative threshold of 0.1 for false discovery rate adjusted p-value. These filters are now described in Supplementary Methods in section 3.10. All relevant R codes (with detailed comments) for detection of differential splicing at level of elementary splicing events are deposited on GitHub (<https://github.com/VGrinev/transcriptome-analysis>, section “Exon-exon junctions”).

On page 13 the authors “used the Cufflinks-assembled transcriptome of Kasumi-1 cells to select all 762 transcripts of the genes affected by RUNX1/RUNX1T1-dependent differential splicing. These genes gave rise to 214 differentially spliced and 548 non-differentially spliced RNA isoforms.” – it is unclear if the authors are referring to 762 or 214 differentially spliced isoforms? What are the 548 non-differentially spliced RNA isoforms?

In the revised manuscript, we clarified this paragraph as follow:

“To further detail above functional findings, we used RNA-Seq data (Supplementary Table 1, GSE54478) and Cufflinks/Cuffdiff algorithm to assemble of full-length RNA molecules as well as to quantify their expression in Kasumi-1 cells. We found that genes affected by RUNX1/RUNX1T1-dependent differential splicing encode 752 transcripts, of which 214 isoforms were differentially spliced. Here and below, the term “differentially spliced isoform” indicates a molecule of RNA with at least one differential exon-exon junction (up- or down-regulated), while a non-differential isoform is a transcript lacking any diffEEJs.”

Furthermore, The authors seem to be cherry picking which method they use to generate gene list; for example page 13 “we used the Cufflinks-assembled transcriptome of Kasumi-1 cells to select all 762 transcripts of the genes affected by RUNX1/RUNX1T1-dependent differential splicing.” while on page 6 “we identified 395 differentially used exons [...] over 275 individual genes” and “ 177 differential EEJs (diffEEJs) [...] over 150 genes”, whereas figure 1 c shows 326 isoforms overlapping between the two detection methods. Again, we understand the need to use different methods, but these need to be explained and properly compared. Are we combining the results of both methods, or only looking at overlap. How many of these events have been validated by RT-PCR?

We used various methods to get a comprehensive picture of differential splicing following RUNX1/RUNX1T1 knockdown, and we do not have really strong arguments in favor of only looking at overlap. In brief, we detected differential splicing at the level of elementary splicing events: differential usage of exons (DEXSeq, limma/diffSplice and JunctionSeq) and differential usage of exon-exon junctions (limma/diffSplice and JunctionSeq). In our opinion, identification of differential splicing at the level of exons or exon-exon junctions is more sensitive and accurate than algorithms based on the assembly of full-length transcripts (fewer steps – less data loss, less algorithmic artifacts and less technical noise). However, this approach has a drawback as well if we want to conduct subsequent functional annotation of such splicing events. We circumvented this limitation in the section “Differential splicing produces proteins with unique conserved domain structures and affects multiple cellular processes” by assembling full-length transcripts and identifying among them those that were differentially spliced (see details in previous comment). It is important to note that we did not use the assembling of full-length transcripts for the primary identification of differential splicing events. We used these transcripts only for predicting the functional consequences of the differential splicing events identified and described in section “Change in the expression of the fusion oncogene RUNX1/RUNX1T1 leads to differential splicing in the leukaemic transcriptome.”

The list of 17 qPCR validated differential splicing events is now included in Supplementary Table 7.

Finally, a major criticism of the used methods is the lack of de novo transcriptome reconstruction, and the reliance on reference transcriptome. Indeed, cancer cells have been shown to express novel isoforms, and these could be absent in the reference transcriptome. This should at least be discussed.

Indeed, we used reference-based Cufflinks suite for assembling of transcriptomes. However, we applied the RABT assembler (Roberts A. et al. Identification of novel transcripts in annotated genomes using RNA-Seq. // Bioinformatics. 2011 Sep 1;27(17):2325-2329) to build novel transcripts upon a known reference annotation. As a result, we found 43094 transcripts in siMM- and/or siRR-treated Kasumi-1 cells with reasonable expression level (FPKM \geq 1). Of these transcripts, 33168 transcripts are identical to references ones (class code "="), 117 transcripts are completely new (class code "-"), 8711 transcripts are potentially novel isoforms that share at least one exon-exon junction (per transcript) with reference transcripts (class code "j"), 383 transcripts are potentially novel isoforms that demonstrate generic exonic overlap with reference transcripts (class code "o"), 67 transcripts include introns that overlap reference introns on the opposite strand(-s) (class code "s"), and 648 transcripts demonstrate exonic overlap(-s) with reference on the opposite strand(-s) (class code "x"). PARL-02 and PARL-02 transcripts (see Figure 7) are examples of new transcripts in our dataset.

A detailed description of the assembly procedure is provided in Supplementary Methods, and respective scripts are deposited on GitHub (<https://github.com/VGrinev/transcriptome-analysis>, section "Transcriptome assembly").

4) All the data is supported by one siRNA in one cell line. The authors should acknowledge this limitation. Also please show whether RUNX1/RUNX1T1 affects cell viability. Differentially splicing changes have been previously associated with cell death, therefore it is critical to show that the changes detected here are not a mere reflection of cell dying.

We have now included additional data obtained with SKNO-1 cells, another t(8;21) cell line representing a more mature phenotype compared to Kasumi-1. In addition, microarray and RNA-Seq data from primary leukemias and normal CD34-positive cells were widely used in this study as well (Supplementary Table 1).

As for siRNAs, we used mismatch siMM as control and siRR targeting the fusion site of the RUNX1/RUNX1T1 mRNA. This focus on the fusion site results in a very small sequence space available for alternative siRNAs. Importantly, the knockdown of RUNX1/RUNX1T1 does not negatively affect cell viability, but causes cell cycle arrest, differentiation and senescence in a strictly RUNX1/RUNX1T1-dependent manner. As a matter of fact, we use siRR routinely as a negative control for cell systems that do not express RUNX1/RUNX1T1 again indicating its high specificity. These siRNAs were repeatedly tested and successfully applied not only by us, but also by multiple other groups working on RUNX1/RUNX1T1.

5) A recent study (Yoshimi et al. Nature 2019.) uncovered combination of mutations in splicing factors, e.g., SRSF2, and other genomic alterations in AML using RNA-seq datasets. Do any of these datasets analyzed here have a mutation in a splicing factor which would explain splicing changes?

Obviously, we are not addressing splicing factor mutations in this study, but focusing on the effect of RUNX1/RUNX1T1 expression on splicing in t(8;21)-positive leukemia. We downloaded publicly available data with all the mutations found in Kasumi-1 and SKNO-1 cells (<https://portals.broadinstitute.org/ccle>). Kasumi-1 cells do not have mutations in splicing factors. SKNO-1 cells harbor mutations in the

following splicing factors: in-frame insertion in *SF3B2* gene (chr11:66052428-66052429), silent mutation in *RBM39* gene (chr20:35725047-35725047), nonsense mutation in *U2AF1L4* gene (chr19:35743880-35743880) and silent mutation in *SRSF2* gene (chr17:76736912-76736912). Mutation in *SF3B2* gene is indel rs146381373 (<https://www.ncbi.nlm.nih.gov/snp/rs146381373>). There are no publications on this mutation and there is no data on its clinical significance in ClinVar. As for *U2AF1L4* gene, this gene is expressed at very low level in Kasumi-1 and SKNO-1 cells (2.7 ± 0.65 RPKM for *U2AF1L4* gene comparing to, for instance, 145.6 ± 21.2 RPKM for *SRSF2* gene in Kasumi-1 cells). With such low expression levels, we do not think that this gene has any significant effect on the phenotype our cell systems.

6) *The authors should be careful when inferring causality. For example on page 5 “These findings strongly support a regulatory role of this leukaemic fusion protein in mRNA splicing and predict changes in splicing pattern in response to perturbing RUNX1/RUNX1T1 activity.” This effect could be due to direct changes in RUNX1/RUNX1T1 binding, as well as indirect for example by changing splicing factor expression. To prove direct binding the authors should prevent/abolish binding at a specific site and demonstrate a change in splicing.*

That is right, and we softened our statement in the revised manuscript.

Also on page 17: *“In conclusion, we have shown here that RUNX1/RUNX1T1 affects alternative splicing of RNA in AML [...] and by modulating expression of splicing components.”- Yet the authors do not detect changes in splicing factors that are likely to regulated the spliced events as stated “However, the expression of the SRSF7 gene itself is not altered after the RUNX1/RUNX1T1 knockdown.”. If the author want to link a specific factor to spliced isoforms detected in this study, they could integrate findings from ENCODE cell lines with splicing factor knockdown, for example SRFS7 KD.*

We thank the reviewer for this suggestion, which encouraged us to include a new data set consisting of combined ENCODE and own functional data. We have incorporated ENCODE data from K562 showing SRSF7 and RBFOX2 binding data for our analysis of TRAPPC2L and PTK2B. Moreover, we included own knockdown experiments investigating the relevance of SRSF7 and RBFOX2 for the alternative splicing of TRAPPC2L and PTK2B. These data are now shown in Figure 10 and Supplementary Figure 6.

7) *The authors perform splicing-factor binding analysis using published motifs, yet they fail to include several relevant high-throughput studies, including Ray et al. Nature 2013, Dominguez et al. Mol Cell 2018 and binding data generated by the ENCODE consortium (www.encode.org and <https://doi.org/10.1101/179648>). Moreover, it is unclear from the material and methods whether control exon/intron sequences were randomly selected from the genome, or selected from known alternatively spliced exons that are not differentially spliced in the studied experiment. Since the sequence composition is quite different between constitutive and alternative exons, this is a critical question.*

There are two critical points: i) motif inference and assessment of motif occurrence in sequence of interest; ii) assessment of the motif importance in determining the class probability (non-differential or differential) of given exon-exon junction. The first point is now described in detail in the section “3.16. Development a list of features associated with EEJs” of Supplementary Methods. In brief, for each splicing protein with experimentally identified binding sites, we performed a motif search by the discriminative motif discovery algorithm against a background set of randomly extracted human intronic and exonic sequences. The primary motif was refined by the function `refinePWMMotif` from the R/Bioconductor library `motifRG` with the default settings and was converted into \log_2 position weight matrix with correction against a background nucleotides frequency. Motif occurrence in a sequence of interest was determined by the function `countPWM` from the R/Bioconductor library `Biostrings` v.2.42.0 and was normalized relative to the length of analysed sequence. We used the 99th quantile of the motif weight distribution as a threshold in the identification of true motif occurrence. All relevant R codes (with detailed comments) concerning this issue are deposited on GitHub (<https://github.com/VGrinev/transcriptome-analysis>, section “Motif analysis”).

The second point is now described in detail in the section “3.16. Development a list of features associated with EEJs” and mainly in the section “3.17. Data mining with the random forest meta-classifier” of Supplementary Methods. In brief, we included in our primary data matrix all non-differential and differential exon-exon junctions identified in the transcriptome of cells of interest. Each exon-exon junction was annotated with sequence, sequence-related, functional, and structural features that were extracted from four types of genomic/RNA elements: 100-bp fragment of the upstream exon, 300-bp fragment from the 5' end of the intron, 300-bp fragment from the 3' end of the intron, and 100-bp fragment of the downstream exon. Additionally, each exon-exon junction was described with a set of nearest epigenetic marks. In total, the complete list of primary features included 1680 items. The importance of each feature (for instance, importance of motif to splicing factor of interest in determining the class probability of given exon-exon junction) was determined by calculating the total decrease in the node impurities from splitting on the feature averaged over all classification trees in random forest. The node impurity was measured with the Gini index. For classification, the primary data matrix was reduced to only important features (according to Gini index). Next, for each run of the random forest meta-classifier, the data matrix was randomly sampled on two sub-matrices: the training set (70% of input matrix) and the test set (30% of input matrix). We used the training set for machine learning, and calculation of the proximity matrix and marginal effects of features. The test set was used to assess the classification accuracy. So, you see we randomly sampled non-differential and differential splicing events, and it is the only correct approach to solving such issues. All relevant R codes (with detailed comments) concerning this point are deposited on GitHub (<https://github.com/VGrinev/transcriptome-analysis>, section “Data mining”).

Additionally, the material and methods states different methods and threshold were used for the detection and scoring of distinct binding motifs. These differences could introduce detection bias which should be discussed in the text. Whether motifs scored by these different methods can be directly compared as done here, remains to be determined.

We evaluated the importance of a particular motif for differential splicing using a random forest meta-classifier. One of the significant advantages of this approach is independence of the result from the scale of feature values. With this approach, we have been able to compare the importance of two features expressed in nominal and quantitative scales.

MINOR POINTS

1) *Please clarify the comment and figure legend on Page 6 “ At the same time, no preference was found among the exons of non-coding RNAs (Fig. 1i).”*

In the revised manuscript, we clarified this phrase as follows: “At the same time, differentially used exons are not over- or under-represented among the exons of non-coding RNAs compared to non-differentially used exons.”

REVIEWERS' COMMENTS

Reviewer #1 (Remarks to the Author):

The authors have substantially improved the manuscript by addressing the reviewers' concerns. The authors have added additional bioinformatics analyses, data sets and functional validation of key findings. The clarity of the manuscript was much improved, now allowing the reader to follow the experimental line. The manuscript improves our understanding of alternative splicing in cancer and how it is also affected by bone fide oncogene fusions.

I have few minor comments:

Line 89: please define "published data sets" in the main text so that the source and content is clear to the reader.

Line 150: qPCR should read qRT-PCR

Line 260: define "up-to-date" in the main text

Line: 332: t(8;21) is mainly associated with AML FAB M2, which is not considered immature.

Line 345: "reported" should read "report"

Reviewer #2 (Remarks to the Author):

The authors have extensively revised the manuscript, added elegant new experiments and addressed multiple questions raised by both reviewers. Findings from this study will advance our understanding of alternative splicing misregulation in tumors, by linking an aberrant transcription factor with splicing defects. However, several questions should still be addressed before the manuscript is ready for publication.

Major points:

1) The author state that "Induction of dCas9-KRAB neither affected PARL-01 expression nor basal levels of PARL-03" – where is that data shown?

2) In figure 8f, there is a difference in colony number at baseline in untreated cells. Therefore, the effect of the BI-D1870 inhibitor is unclear and the lack of P-values makes the interpretation of the data difficult.

3) The authors state that "19 [splicing factors] including USB1 are associated with RUNX1/RUNX1T1 binding sites implying them as direct target genes (Fig. 9c)". Please provide state what the remaining 18 are? Are RBFOX2 and SRSF7 also targets of RUNX1/RUNX1T1?

4) The authors state "Motifs for all splicing factors of this list similarly affected the discrimination between differential versus non differential splicing events (data not shown)." Given that the author will focus on RBFOX2 in Fig.10, please add the data for RBFOX2 in Fig. 9e.

5) The first result section and related figures are basically a catalog of number of splicing events, and could be shorten and partially moved to supplemental data. Several of the supplementary tables are long list of complete data that need to be digested for the reader in some graphical form. For example, Table S2 says "Functional annotation by DAVID uncovered a highly significant enrichment of RUNX1/RUNX1T1 binding at gene loci that are subject to alternative splicing and splice variants", yet this is not obvious in the table. Similarly, the author state "Furthermore, we observed that RUNX1/RUNX1T1 binding sites are often present within genes encoding splicing associated factors or in their immediate vicinity (Supplementary Table 3)."; the cited table is a complete list of direct target genes of RUNX1/RUNX1T1 fusion protein in siMM-treated Kasumi-1 cells – is the reader supposed to extract manually the splicing associated factors or their

immediate vicinity? Considering there are >1000 RNA-binding proteins, what is the likelihood that RUNX1/RUNX1T1 binds near to one.

Minor points:

- 1) While we acknowledge the authors have extensively revised the text, we find the article still extremely complicated and difficult to read. Improving the clarity of the figures and further reducing acronyms, as well as additional text edits would greatly enhance the paper.
- 2) Please show the magnitude of SRSF7-KD in Fig S7.

We again found the reviewer's comments very helpful and have addressed them as below:

Reviewer #1 (Remarks to the Author):

The authors have substantially improved the manuscript by addressing the reviewers' concerns. The authors have added additional bioinformatics analyses, data sets and functional validation of key findings. The clarity of the manuscript was much improved, now allowing the reader to follow the experimental line. The manuscript improves our understanding of alternative splicing in cancer and how it is also affected by bone fide oncogene fusions.

We are very grateful about this very positive general comment.

Line 89: please define "published data sets" in the main text so that the source and content is clear to the reader.

We have added this information. The text reads now: "To detect a potential association between RUNX1/RUNX1T1 and RNA splicing, we initially analyzed published RNA-seq, ChIP-seq and DNase-hypersensitive-site-seq (DHS-seq) data sets for links between RUNX1/RUNX1T1 and RNA processing (Supplementary Data 1)."

Line 150: qPCR should read qRT-PCR
Changed

Line 260: define "up-to-date" in the main text
We have included release dates.

Line: 332: t(8;21) is mainly associated with AML FAB M2, which is not considered immature.
"Immature" was deleted.

Line 345: "reported" should read "report"
We think the reviewer meant "resulted" and "result". We have changed it accordingly.

Reviewer #2 (Remarks to the Author):

The authors have extensively revised the manuscript, added elegant new experiments and addressed multiple questions raised by both reviewers. Findings from this study will advance our understanding of alternative splicing misregulation in tumors, by linking an aberrant transcription factor with splicing defects. However, several questions should still be addressed before the manuscript is ready for publication.

Again, we are very grateful about this very positive general comment by reviewer 2.

1) The author state that “Induction of dCas9-KRAB neither affected PARL-01 expression nor basal levels of PARL-03” – where is that data shown?

We have rephrased this statement to: “Induction of dCas9-KRAB did not affect PARL-01 expression with 0.98 and 1.01fold changes for siMM and siRR-treated cells, respectively.

2) In figure 8f, there is a difference in colony number at baseline in untreated cells. Therefore, the effect of the BI-D1870 inhibitor is unclear and the lack of P-values makes the interpretation of the data difficult.

This difference is caused by the knockdown of RUNX1/ETO that diminishes clonogenicity on its own. However, combination of knockdown with inhibitor causes a significantly stronger loss of clonogenicity. For better comparison, we have now included the p values into the legend.

3) The authors state that “19 [splicing factors] including USB1 are associated with RUNX1/RUNX1T1 binding sites implying them as direct target genes (Fig. 9c).”. Please provide state what the remaining 18 are? Are RBFOX2 and SRSF7 also targets of RUNX1/RUNX1T1?

19 was a spelling mistake, there are only 16 factors mentioned in the manuscript. We have corrected this mistake. The 16 factors are listed in supplemental data 8; we have added this information in the text. RBFOX2 and SRSF7 do not harbour substantial RUNX1/RUNX1T1 peaks and are not considered to be direct target genes.

4) The authors state “Motifs for all splicing factors of this list similarly affected the discrimination between differential versus non differential splicing events (data not shown).” Given that the author will focus on RBFOX2 in Fig.10, please add the data for RBFOX2 in Fig. 9e.

We selected RBFOX2 for further experimental validation because the expression of this gene was down-regulated following RUNX1/RUNX1T1 knockdown. Moreover, we found RBFOX2 binding site near of exon E23 of PTK2B. However, globally there are no statistically significant differences between non-diffEEJs and diffEEJs (see Supplementary Table "1) in terms of RBFOX2 binding sites frequency or affinity. Thus, we think that inclusion of such a figure does not add any value to the manuscript.

5) The first result section and related figures are basically a catalog of number of splicing events, and could be shorten and partially moved to supplemental data. Several of the supplementary tables are long list of complete data that need to be digested for the reader in some graphical form. For example, Table S2 says “Functional annotation by DAVID uncovered a highly significant enrichment of RUNX1/RUNX1T1 binding at gene loci that are subject to alternative splicing and splice variants”, yet this is not obvious in the table.

We have simplified Supplementary Table 3 by removing UCSC-TFBS and SMART enrichments with neither of them being of direct relevance for this study. Both terms are on top of the significantly enriched terms with “alternative splicing” being second and “splice variants” being fourth when sorted for significance.

Similarly, the author state “Furthermore, we observed that RUNX1/RUNX1T1 binding sites are often present within genes encoding splicing associated factors or in their immediate vicinity (Supplementary Table 3).”; the cited table is a complete list of direct target genes of RUNX1/RUNX1T1 fusion protein in siMM-treated Kasumi-1 cells – is the reader supposed to extract manually the splicing associated factors or their immediate vicinity?

Considering there are >1000 RNA-binding proteins, what is the likelihood that RUNX1/RUNX1T1 binds near to one.

In the revised text (lane 393), we are now also referring to Supplemental Data 8, which lists the differentially expressed splicing factor and mRNA surveillance genes and their distance to the nearest RUNX1/RUNX1T1 binding site.

1) While we acknowledge the authors have extensively revised the text, we find the article still extremely complicated and difficult to read. Improving the clarity of the figures and further reducing acronyms, as well as additional text edits would greatly enhance the paper.

We have further edited the text for simplicity and readability. However, we do note that reviewer 1 was satisfied with the changes we already introduced in the first revision.

2) Please show the magnitude of SRSF7-KD in Fig S7.

This information is now included as Supplementary Figure 7d.